# OUT-OF-DISTRIBUTION DETECTION WITH IMPLICIT OUTLIER TRANSFORMATION

**Qizhou Wang**[1]   **Junjie Ye**[2][†]   **Feng Liu**[3]   **Quanyu Dai**[2]   **Marcus Kalander**[2]
**Tongliang Liu**[4]   **Jianye Hao**[2]   **Bo Han**[1][†]
[1]Hong Kong Baptist University   [2]Huawei Noah's Ark Lab
[3]The University of Melbourne   [4] Sydney AI Centre, The University of Sydney
{csqzwang, bhanml}@comp.hkbu.edu.hk   fengliu.ml@gmail.com   tongliang.liu@sydney.edu.au
{yejunjie4, daiquanyu, marcus.kalander, haojianye}@huawei.com

## ABSTRACT

*Outlier exposure* (OE) is powerful in *out-of-distribution* (OOD) detection, enhancing detection capability via model fine-tuning with surrogate OOD data. However, surrogate data typically deviate from test OOD data. Thus, the performance of OE, when facing unseen OOD data, can be weakened. To address this issue, we propose a novel OE-based approach that makes the model perform well for unseen OOD situations, even for unseen OOD cases. It leads to a min-max learning scheme—searching to synthesize OOD data that leads to worst judgments and learning from such OOD data for uniform performance in OOD detection. In our realization, these worst OOD data are synthesized by transforming original surrogate ones. Specifically, the associated transform functions are learned *implicitly* based on our novel insight that model perturbation leads to data transformation. Our methodology offers an efficient way of synthesizing OOD data, which can further benefit the detection model, besides the surrogate OOD data. We conduct extensive experiments under various OOD detection setups, demonstrating the effectiveness of our method against its advanced counterparts. The code is publicly available at: github.com/qizhouwang/doe.

## 1 INTRODUCTION

Deep learning systems in the open world often encounter *out-of-distribution* (OOD) data whose label space is disjoint with that of the *in-distribution* (ID) samples. For many safety-critical applications, deep models should make reliable predictions for ID data, while OOD cases (Bulusu et al., 2020) should be reported as anomalies. It leads to the well-known OOD detection problem (Lee et al., 2018c; Fang et al., 2022), which has attracted intensive attention in reliable machine learning.

OOD detection remains non-trivial since deep models can be over-confident when facing OOD data (Nguyen et al., 2015; Bendale & Boult, 2016), and many efforts have been made in pursuing reliable detection models (Yang et al., 2021; Salehi et al., 2021). Building upon discriminative models, existing OOD detection methods can generally be attributed to two categories, namely, *post-hoc* approaches and *fine-tuning* approaches. The post-hoc approaches assume a well-trained model on ID data with its fixed parameters, using model responses to devise various *scoring functions* to indicate ID and OOD cases (Hendrycks & Gimpel, 2017; Liang et al., 2018; Lee et al., 2018c; Liu et al., 2020; Sun et al., 2021; 2022; Wang et al., 2022). By contrast, the fine-tuning methods allow the target model to be further adjusted, boosting its detection capability by regularization (Lee et al., 2018a; Hendrycks et al., 2019; Tack et al., 2020; Mohseni et al., 2020; Sehwag et al., 2021; Chen et al., 2021; Du et al., 2022; Ming et al., 2022; Bitterwolf et al., 2022). Typically, fine-tuning approaches benefit from explicit knowledge of unknowns during training and thus generally reveal reliable performance across various real-world situations (Yang et al., 2021).

For the fine-tuning approaches, *outlier exposure* (OE) (Hendrycks et al., 2019) is among the most potent ones, engaging surrogate OOD data during training to discern ID and OOD patterns. By mak-

---

[†]Correspondence to Bo Han (bhanml@comp.hkbu.edu.hk) and Junjie Ye (yejunjie4@huawei.com).

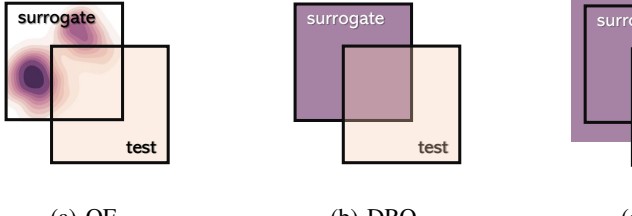

|          |          |          |
|:--------:|:--------:|:--------:|
| (a) OE   | (b) DRO  | (c) DOE  |

Figure 1: Comparison between OE, DRO, and DOE. Black boxes indicate support sets for surrogate/test OOD data. Intensities of color indicate the coverage of learning schemes—a deeper colored region indicates the associated model can make more reliable detection therein. As we can see, OE directly makes the model learn from surrogate OOD data, largely deviating from test OOD situations. DRO further makes the model perform uniformly well regarding sub-populations, and the model can excel in the support set of the surrogate case. Moreover, DOE makes the model learn from additional OOD data besides surrogate cases, covering wider OOD situations (exceeding the support set) than that of OE and DRO. Thus, OOD detection capability increases from left to right.

ing these surrogate OOD data with low-confident predictions, OE explicitly enables the detection model to learn knowledge for effective OOD detection. A caveat is that one can hardly know what kind of OOD data will be encountered when the model is deployed. Thus, the distribution gap exists between surrogate (training-time) and unseen (test-time) OOD cases. Basically, this distribution gap is harmful for OOD detection since one can hardly ensure the model performance when facing OOD data that largely deviate from the surrogate OOD data (Yang et al., 2021; Dong et al., 2020).

Addressing the OOD distribution gap issue is essential but challenging for OE. Several works are related to this problem, typically shrinking the gap by making the model learn from additional OOD data. For example, Lee et al. (2018a) synthesize OOD data that the model will make mistakes by generative models, and the synthetic data are learned by the detection model for low confidence predictions. However, synthesizing unseen is intractable in general (Du et al., 2022), meaning that corresponding data may not fully benefit OE training. Instead, Zhang et al. (2023) mixup ID and surrogate OOD data to expand the coverage of OOD cases; and Du et al. (2022) sample OOD data from the low-likelihood region of the class-conditional distribution in the low-dimensional feature space. However, linear interpolation in the former can hardly cover diverse OOD situations, and feature space data generation in the latter may fail to fully benefit the underlying feature extractors. Hence, there is still a long way to go to address the OOD distribution gap issue in OE.

To overcome the above drawbacks, we suggest a simple yet powerful way to access extra OOD data, where we transform available surrogate data into new OOD data that further benefit our detection models. The key insight is that model perturbation implicitly leads to data transformation, and the detection models can learn from such *implicit data* by model updating after its perturbation. The associated transform functions are free from tedious manual designs (Zhang et al., 2023; Huang et al., 2023) and complex generative models (Lee et al., 2018b) while remaining flexible for synthetic OOD data that deviate from original data. Here, two factors support the effectiveness of our data synthesis: 1) implicit data follow different distribution from that of the original one (cf., Theorem 1) and 2) the discrepancy between original and transformed data distributions can be very large, given that our detection model is deep enough (cf., Lemma 1). It indicates that one can effectively synthesize extra OOD data that are largely different from the original ones. Then, we can learn from such data to further benefit the detection model.

Accordingly, we propose *Distributional-agnostic Outlier Exposure* (DOE), a novel OE-based approach built upon our implicit data transformation. The "distributional-agnostic" reflects our ultimate goal of making the detection models perform uniformly well with respect to various unseen OOD distributions, accessing only ID and surrogate OOD data during training. In DOE, we measure the model performance in OOD detection by the *worst OOD regret* (WOR) regarding a candidate set of OOD distributions (cf., Definition 2), leading to a min-max learning scheme as in equation 6. Then, based on our systematic way of implicit data synthesis, we iterate between 1) searching implicit OOD data that lead to large WOR via model perturbation and 2) learning from such data for uniform detection power for the detection model.

DOE is related to distributionally robust optimization (DRO) (Rahimian & Mehrotra, 2019), which similarly learns from the worst-case distributions. Their conceptual comparison is summarized in Figure 1. Therein, DRO considers a close-world setting, striving for uniform performance regarding various data distributions in the support (Sagawa et al., 2020). However, it fails in the open-world OOD settings that require detecting unseen data (cf., Section 5.3), which is the part of the test support that is disjoint with the surrogate one in Figure 1(b). By contrast, our data transformation offers an effective approach in learning from unseen data, considering the region's uniform performance beyond the support. Thus, DOE can mitigate the distribution gap issue to some extent, reflected by the smaller disjoint region than the DRO case in Figure 1(c).

We conduct extensive experiments in Section 5 on widely used benchmark datasets, verifying the effectiveness of our method with respect to a wide range of different OOD detection setups. For common OOD detection, our DOE reduces the average FPR95 by $7.26\%$, $20.30\%$, and $13.97\%$ compared with the original OE on CIFAR-10, CIFAR-100, and ImageNet datasets. For hard OOD detection, our DOE reduces the FPR95 by $7.45\%$, $7.75\%$, and $4.09\%$ compared with advanced methods regarding various hard OOD datasets.

## 2 PRELIMINARY

Let $\mathcal{X} \subseteq \mathbb{R}^d$ denote the input space and $\mathcal{Y} = \{1, \ldots, C\}$ the label space. We consider the ID distribution $D_{\mathrm{ID}}$ defined over $\mathcal{X} \times \mathcal{Y}$ and the OOD distribution $D_{\mathrm{OOD}}$ defined over $\mathcal{X}$. In general, the OOD distribution $D_{\mathrm{OOD}}$ is defined as an irrelevant distribution whose label set has no intersection with $\mathcal{Y}$ (Yang et al., 2021), which is unseen during training and should not be predicted by the model.

### 2.1 SOFTMAX SCORING

Building upon the model $h \in \mathcal{H} : \mathcal{X} \to \mathbb{R}^C$ with logit outputs, our goal is to utilize the *scoring function* $s : \mathcal{X} \to \mathbb{R}$ in discerning test-time inputs given by $D_{\mathrm{ID}}$ from that of $D_{\mathrm{OOD}}$. Typically, if the score value $s(\boldsymbol{x})$ is greater than a threshold $\tau \in \mathbb{R}$, the associated input $\boldsymbol{x} \in \mathcal{X}$ is taken as an ID case, otherwise an OOD case. A representative scoring function in the literature is the maximum softmax prediction (MSP) (Hendrycks & Gimpel, 2017), following

$$s_{\mathrm{MSP}}(\boldsymbol{x}; h) = \max_k \texttt{softmax}_k h(\boldsymbol{x}), \tag{1}$$

where $\texttt{softmax}_k(\cdot)$ denotes the $k$-th element of a softmax output. Since the true labels of OOD are not in the label space, the model will return lower scores for them than ID cases in expectation.

### 2.2 OUTLIER EXPOSURE

Unfortunately, for a normally trained model $h(\cdot)$, MSP may make over-confident predictions for some OOD data (Liu et al., 2020), which is detrimental in effective OOD detection. To this end, OE (Hendrycks et al., 2019) boosts the detection capability by making the model $h(\cdot)$ learn from the surrogate OOD distribution $D_{\mathrm{OOD}}^{\mathrm{s}}$, with the associated learning objective of the form:

$$\mathcal{L}(h) = \underbrace{\mathbb{E}_{D_{\mathrm{ID}}} \left[ \ell_{\mathrm{CE}}(h(\boldsymbol{x}), y) \right]}_{\mathcal{L}_{\mathrm{CE}}(h; D_{\mathrm{ID}})} + \lambda \underbrace{\mathbb{E}_{D_{\mathrm{OOD}}^{\mathrm{s}}} \left[ \ell_{\mathrm{OE}}(h(\boldsymbol{x})) \right]}_{\mathcal{L}_{\mathrm{OE}}(h; D_{\mathrm{OOD}}^{\mathrm{s}})}, \tag{2}$$

where $\lambda$ is the trade-off parameter, $\ell_{\mathrm{CE}}(\cdot)$ is the cross-entropy loss, and $\ell_{\mathrm{OE}}(\cdot)$ is defined by Kullback-Leibler divergence to the uniform distribution, which can be written as $\ell_{\mathrm{OE}}(h(\boldsymbol{x})) = -\sum_k \texttt{softmax}_k h(\boldsymbol{x})/C$. Basically, the OE loss $\ell_{\mathrm{OE}}(\cdot)$ plays the role of regularization, making the model learn from surrogate OOD data with low confident predictions. Since the model can see some OOD data during training, OE typically reveals reliable performance in OOD detection.

Note that since we know nothing about unseen during training, the surrogate distribution $D_{\mathrm{OOD}}^{\mathrm{s}}$ is largely different from the real one $D_{\mathrm{OOD}}$ in general. Then, the difference between surrogate and unseen OOD data leads to the OOD distribution gap between training- (i.e., $D_{\mathrm{OOD}}^{\mathrm{s}}$) and test-time (i.e., $D_{\mathrm{OOD}}$) situations. When deployed, the model inherits this data bias, potentially making over-confident predictions for unseen OOD data that differ from the surrogate ones.

## 3 OOD SYNTHESIS

The OOD distribution gap issue stems from our insufficient knowledge about (test-time) unseen OOD data. Therefore, a direct approach is to give the model access to extra OOD data via data synthesis, doing our best to fill the distribution gap between training- and test-time situations.

When it comes to data synthesis, a direct approach is to utilize generative models (Lee et al., 2018a), while generating unseen is intractable in general (Du et al., 2022). Therefore, MixOE (Zhang et al., 2023) mixup ID and surrogate OOD to expand the coverage of various OOD situations, and VOS (Du et al., 2022) generates additional OOD in the embedding space with respect to low-likelihood ID regions. However, the former relies on manually designed synthesizing procedures, which can hardly cover diverse OOD situations. The latter generates OOD in low-dimensional space, which relies on specific assumptions for ID distribution (e.g., a mixture of Gaussian) and hardly benefits the underlying feature extractors to learn meaningful OOD patterns.

### 3.1 MODEL PERTURBATION FOR DATA SYNTHESIS

Considering previous drawbacks in OOD synthesis, we suggest a new way to access additional OOD data, which is simple yet powerful. Overall, we transform the available surrogate OOD data to synthesize new data that can further benefit our model. The associated transform function is parasitic on our detection model, which is learnable without auxiliary deep models or manual designs.

The key insight is that perturbing model parameters have the same impact as transforming data, where specific model perturbations indicate specific transform functions. For the beneficial data of our interest (e.g., the worst OOD data), we can implicitly get them access by finding the corresponding model perturbation. Updating the detection model thereafter, it can learn from the transformed data (i.e., the beneficial ones) instead of the original inputs. Now, we formalize our intuition.

We study the piecewise affine ReLU network model (Arora et al., 2018), covering a large group of deep models with ReLU activations, fully connected layers, convolutional layers, residual layers, etc. Here, we consider the recursive definition of a $L$-layer ReLU network, following

$$\boldsymbol{z}^{(l)} = h^{(l)}(W^{(l-1)}\boldsymbol{z}^{(l-1)}) \text{ for } l = 1, \ldots, L, \tag{3}$$

where $W^{(l)} \in \mathbb{R}^{n_l \times n_{l-1}}$ is the $l$-th layer weights and $h^{(l)}(\boldsymbol{z}) = \max\{0, t\}$ the ReLU activation. We have $\boldsymbol{z}^{(1)} = \boldsymbol{x}$ the model input and $\boldsymbol{z}^{(L)} = h(\boldsymbol{x})$ the model output. If necessary, we write $h_W$ in place of $h$ with the joint form of weights $W = \{W^{(l)}\}_{l=1}^{L}$ that contains all trainable parameters.

Our discussion is on a specific form of model perturbation named *multiplicative perturbation*.

**Definition 1** (**Multiplicative Perturbation** (Petzka et al., 2021)). *For a L-layer ReLU network $h(\cdot)$, its l-th layer is multiplicatively perturbed if $W^{(l)}$ is changed into*

$$W^{(l)}(I + \alpha A^{(l)}), \tag{4}$$

*where $\alpha > 0$ is the perturbation strength and $A^{(l)} \in \mathbb{R}^{n_{l-1} \times n_{l-1}}$ is the perturbation matrix. Furthermore, the model $h(\cdot)$ is multiplicatively perturbed if all its layers are multiplicatively perturbed.*

Now, we link the multiplicative perturbation of the $l$-th layer to data transformation in the associated embedding space, summarized by the following proposition.

**Proposition 1.** *Considering the data distribution $D$ and the multiplicative perturbation regarding the l-th layer of a ReLU network. Then, measuring in the feature space, multiplicative perturbation is equivalent to data transformation. Further, the transformed data follows a new distribution $D'$ that is different from $D$ if the eigenvalues of $A^{(l)}$ are greater than $0$.*

Therefore, model perturbation offers an alternative way to modify data and their distribution implicitly. Now, we generalize Proposition 1 for the multiplicative perturbation of the model, showing that it can modify the data distribution in the original input space.

**Theorem 1.** *Considering the data distribution $D$ and an $L$-layer ReLU network. Measuring in the input space $\mathcal{X} \subseteq \mathbb{R}^d$, multiplicative perturbation of the model is equivalent to data transformation in the input space following distribution $D'$. Then, $D'$ and $D$ are different if the eigenvalues of $A^{(l)}$ are greater than $0$ and $W^{(l),\dagger} = W^{(l),-1}$ for $l = 1, \ldots, L$.*

The proof of the above theorem directly leads to the following lemma, indicating that our data-synthesizing approach can benefit from the layer-wise architectures of deep models.

**Lemma 1.** *Considering a L-layer ReLU network with the multiplicative perturbation $\{A_L^{(l)}\}_{l=1}^{L}$ and the associated transformed distribution $D'_L$. Then, there exists a $L+1$-layer ReLU network with the multiplicative perturbation $\{A_{L+1}^{(l)}\}_{l=1}^{L+1}$ and the associated transformed distribution $D'_{L+1}$, such that the difference between $D'_{L+1}$ and $D$ is no smaller than the difference between $D'_L$ and $D$.*

All the above proofs can be found in Appendix A, revealing that model perturbation leads to data transformation. There are two points worth emphasizing. First, the distribution of transformed data can be very different from that of the original data under the mild condition of non-negative eigenvalues. Further, the corresponding transform function is complex enough with layer-wise non-linearity, where deep models induce strong forms of transformations (regarding distributions).

## 4    DISTRIBUTIONAL-AGNOSTIC OUTLIER EXPOSURE

Our data synthesis scheme allows the model $h(\cdot)$ to learn from additional OOD data besides the surrogate ones. Recalling that, we aim for the model to perform uniformly well for various unseen OOD data. Then, a critical issue is what kinds of synthesized OOD can benefit our model the most.

To begin with, we measure the detection capability by the worst-case OOD performance of the detection model, leading to the following definition of the *worst OOD regret* (WOR).

**Definition 2** (**Worst OOD Regret**). *For the detection model $h(\cdot)$, its worst OOD regret is*

$$WOR(h) = \sup_{D \in \mathcal{D}_{OOD}} \left[ \mathcal{L}_{OE}(h; D) - \inf_{h' \in \mathcal{H}} \mathcal{L}_{OE}(h'; D) \right], \tag{5}$$

*where $\mathcal{D}_{OOD}$ denotes the set of all OOD distributions and $\mathcal{H}$ is the hypothesis space.*

Minimizing the WOR upper bounds the uniform performance of the detection model for the OOD cases. Therefore, synthetic OOD data that lead to WOR are of our interest, and learning from such data can benefit our model the most. Note that we can also measure the detection capability by the risk, i.e., $\sup_{D \in \mathcal{D}_{OOD}} \mathcal{L}_{OE}(h; D)$, while we find that our regret-based measurement is better since it further considers the fitting power of the model when facing extremely large space of unseen data.

### 4.1    LEARNING OBJECTIVE

The WOR measures the worst OOD regret with respect to the worst OOD distribution, suitable for our perturbation-based data transformation that can lead to new data distributions (cf., Theorem 1). Therefore, to empirically upper-bound the WOR, one can first find the model perturbation that leads to large OOD regret and then update model parameters after its perturbation. Here, an implicit assumption is that the associated data given by model perturbation (with surrogate OOD inputs) are valid OOD cases. It is reasonable since the WOR in equation 5 does not involve any term to make the associated data close to ID data in either semantics or stylish.

Then, we propose an OE-based method for uniformly well OOD detection, namely, *Distributional-agnostic Outlier Exposure* (DOE). It is formalized by a min-max learning problem, namely,

$$\mathcal{L}_{\text{DOE}}(h_W; D_{\text{ID}}, D_{\text{OOD}}^{\text{s}}) = \mathcal{L}_{\text{CE}}(h_W; D_{\text{ID}}) +$$
$$\lambda \underbrace{\max_{P:||P|| \leq 1} \left[ \mathcal{L}_{\text{OE}}(h_{W+\alpha P}; D_{\text{OOD}}^{\text{s}}) - \min_{W'} \mathcal{L}_{\text{OE}}(h_{W'+\alpha P}; D_{\text{OOD}}^{\text{s}}) \right]}_{\text{WOR}_{\text{P}}(h_W; D_{\text{OOD}}^{\text{s}})}, \tag{6}$$

where $\text{WOR}_{\text{P}}(h_W; D_{\text{OOD}}^{\text{s}})$ is a perturbation-based realization for the WOR calculation. Several points therein require our attention. First, ID data remain the same during training and testing, and the distribution gap occurs only for OOD cases. Therefore, WOR is applied only to the surrogate OOD data, and the original risk $\mathcal{L}_{\text{CE}}(h_W; D_{\text{ID}})$ is applied for the ID data. Furthermore, we adopt the implicit data transformation to search for the worst OOD distribution, substituting the search space of distribution $\mathcal{D}_{\text{OOD}}$ by the search space of the perturbation, i.e., $\{P : ||P|| \leq 1\}$. Here, we adopt

a fixed threshold of 1 since one can change the perturbation strength via the parameter $\alpha$. Finally, we adopt the additive perturbation $W + \alpha P$ which is easier to implement than the multiplicative counterpart, and they are equivalent when assuming $P = WA$.

## 4.2 REALIZATION

We consider a stochastic realization of DOE, where ID and OOD mini-batches are randomly sampled in each iteration, denoted by $B_{ID}$ and $B_{OOD}^s$, respectively. The overall DOE algorithm is summarized in Appendix B. Here, we emphasize several vital points.

**Regret Estimation.** The exact regret computation is hard since we need to find the optimal risk for each candidate perturbation. As its effective estimation, following (Arjovsky et al., 2019; Agarwal & Zhang, 2022), we calculate the norm of the gradients with respect to the risk $\mathcal{L}_{OE}$, namely,

$$\text{WOR}_{\text{G}}(h_W; B_{OOD}^s) = ||\nabla_{\sigma|\sigma=1.0}\mathcal{L}_{OE}(\sigma \cdot h_{W+\alpha P}; B_{OOD}^s)||^2. \tag{7}$$

Intuitively, a large value of the gradient norm indicates that the current model is far from optimal, and thus the corresponding regret should be large. It leads to an efficient indicator of regret.

**Perturbation Estimation.** The gradient ascent is employed to find the proper perturbation $P$ for the max operation in equation 6. In each step, the perturbation is updated by

$$P \leftarrow \nabla_P \text{WOR}_{\text{G}}(h_{W+\alpha P}; B_{OOD}^s), \tag{8}$$

with $P$ initialized to 0. We further normalize $P$ using $P_{\text{NORM}} = \text{NORM}(P)$ to satisfy the norm constraint. By default, we employ one step of gradient update as an efficient estimation for its value, which can be taken as the solution for the first-order Taylor approximated model.

**Stable Estimation.** Equation 8 is calculated for the mini-batch of OOD samples, biased from the exact solution of $P$ that leads to the worst regret regarding the whole training sample. To mitigate the gap, for the resultant $P_{\text{NORM}}$, we adopt its moving average across training steps, namely,

$$P_{\text{MA}} \leftarrow (1 - \beta)P_{\text{MA}} + \beta P_{\text{NORM}}, \tag{9}$$

where $\beta \in (0, 1]$ is the smoothing strength. Overall, a smaller $\beta$ indicates that we take the average for a wider range of steps, leading to a more stable estimation of the perturbation.

**Scoring Function.** After training, we adopt the MaxLogit scoring (Hendrycks et al., 2022) in OOD detection, which is better than the MSP scoring when facing large semantic spaces. It is of the form:

$$s_{\text{ML}}(\boldsymbol{x}; h) = \max_k h_k(\boldsymbol{x}), \tag{10}$$

where $h_k(\cdot)$ denotes the $k$-th element of the logit output. In general, a large value of $s_{\text{ML}}(\boldsymbol{x}; h)$ indicates the high confidence of the associated $\boldsymbol{x}$ to be an ID case.

## 5 EXPERIMENTS

This section conducts extensive experiments in OOD detection. In Section 5.1, we verify the superiority of our DOE against state-of-the-art methods on both the CIFAR (Krizhevsky & Hinton, 2009) and the ImageNet (Deng et al., 2009) benchmarks. In Section 5.2, we demonstrate the effectiveness of our method for hard OOD detection. In Section 5.3, we further conduct an ablation study to understand our learning mechanism in depth. The code is publicly available at: github.com/qizhouwang/doe.

**Baseline Methods.** We compare our DOE with advanced methods in OOD detection. For post-hoc approaches, we consider MSP (Hendrycks & Gimpel, 2017), ODIN (Liang et al., 2018), Mahalanobis (Lee et al., 2018c), Free Energy (Liu et al., 2020), ReAct (Sun et al., 2021), and KNN (Sun et al., 2022); for fine-tuning approaches, we consider OE (Hendrycks et al., 2019), CSI (Tack et al., 2020), SSD+ (Sehwag et al., 2021), MixOE (Zhang et al., 2023), and VOS (Du et al., 2022).

**Evaluation Metrics.** The OOD detection performance of a detection model is evaluated via two representative metrics, which are both threshold-independent (Davis & Goadrich, 2006): the false positive rate of OOD data when the true positive rate of ID data is at $95\%$ (FPR95); and the *area*

Table 1: Comparison in OOD detection on the CIFAR and ImageNet benchmarks. ↓ (or ↑) indicates smaller (or larger) values are preferred; a bold font indicates the best results in a column.

| Methods | CIFAR-10 | | CIFAR-100 | | ImageNet | |
|---|---|---|---|---|---|---|
| | FPR95 ↓ | AUROC ↑ | FPR95 ↓ | AUROC ↑ | FPR95 ↓ | AUROC ↑ |
| Post-hoc Approaches | | | | | | |
| MSP | 53.77 | 88.40 | 76.73 | 76.24 | 75.32 | 76.96 |
| ODIN | 42.80 | 88.69 | 63.25 | 75.72 | 77.43 | 71.04 |
| Mahalanobis | 34.98 | 93.21 | 65.57 | 78.03 | 86.50 | 58.78 |
| Free Energy | 37.77 | 88.27 | 71.56 | 78.51 | 71.14 | 79.50 |
| ReAct | 58.22 | 82.21 | 69.94 | 78.21 | 70.31 | 81.42 |
| KNN | 34.56 | 93.43 | 50.24 | 86.73 | 64.75 | 80.91 |
| Fine-tuning Approaches | | | | | | |
| OE | 12.41 | 97.85 | 45.68 | 87.61 | 73.80 | 78.90 |
| CSI | 17.39 | 96.87 | 83.72 | 65.94 | 86.80 | 65.54 |
| SSD+ | 14.84 | 97.36 | 56.65 | 87.38 | 64.55 | 77.46 |
| MixOE | 13.55 | 97.59 | 52.04 | 86.46 | 74.36 | 77.28 |
| VOS | 31.55 | 91.56 | 73.43 | 79.98 | 87.87 | 61.36 |
| DOE | **5.15** | **98.78** | **25.38** | **93.97** | **59.83** | **83.54** |

*under the receiver operating characteristic curve* (AUROC), which can be viewed as the probability of the ID case having greater score than that of the OOD case.

**Pre-training Setups.** For the CIFAR benchmarks, we employ the WRN-40-2 (Zagoruyko & Komodakis, 2016) as the backbone model following (Liu et al., 2020). The models have been trained for 200 epochs via empirical risk minimization, with a batch size 64, momentum 0.9, and initial learning rate 0.1. The learning rate is divided by 10 after 100 and 150 epochs. For the ImageNet, we employ ResNet-50 (He et al., 2016) with well-trained parameters downloaded from the PyTorch repository following (Sun et al., 2021).

**DOE Setups.** Hyper-parameters are chosen based on the OOD detection performance on validation datasets, which are separated from ID and surrogate OOD data. For the CIFAR benchmarks, DOE is run for 10 epochs with an initial learning rate of 0.01 and the cosine decay (Loshchilov & Hutter, 2017). The batch size is 128 for ID cases and 256 for OOD cases. The number of warm-up epochs is set to 5. $\lambda$ is 1 and $\beta$ is 0.6. For the ImageNet dataset, DOE is run for 4 epochs with an initial learning rate of 0.0001 and cosine decay. The batch sizes are 64 for both ID and surrogate OOD cases. The number of warm-up epochs is 2. $\lambda$ is 1 and $\beta$ is 0.1. For both the CIFAR and the ImageNet benchmarks, $\sigma$ is uniformly sampled from $\{1e^{-1}, 1e^{-2}, 1e^{-3}, 1e^{-4}\}$ in each training step, which allows covering a wider range of OOD situations than assigning fixed values. Furthermore, the perturbation step is fixed to be 1.

**Surrogate OOD datasets.** For the CIFAR benchmarks, we adopt the tinyImageNet dataset (Le & Yang, 2015) as the surrogate OOD dataset for training. For the ImageNet dataset, we employ the ImageNet-21K-P dataset (Ridnik et al., 2021), which makes invalid classes cleansing and image resizing compared with the original ImageNet-21K (Deng et al., 2009).

## 5.1 COMMON OOD DETECTION

We begin with our main experiments on the CIFAR and ImageNet benchmarks. Model performance is tested on several common OOD datasets widely adopted in the literature (Sun et al., 2022). For the CIFAR cases, we employed Texture (Cimpoi et al., 2014), SVHN (Netzer et al., 2011), Places365 (Zhou et al., 2018), LSUN-Crop (Yu et al., 2015), and iSUN (Xu et al., 2015); for the ImageNet case, we employed iNaturalist (Horn et al., 2018), SUN (Xu et al., 2015), Places365 (Zhou et al., 2018), and Texture (Cimpoi et al., 2014). In Table 1, we report the average performance (i.e., FPR95 and AUROC) regarding the OOD datasets mentioned above. Please refer to Tables 4-5 and 8 in Appendix C for the detailed results.

**CIFAR Benchmarks.** Overall, the fine-tuning methods can lead to effective OOD detection in that they (e.g., OE and DOE) generally demonstrate better results than most of the post-hoc approaches

Table 2: Comparison of DOE and advanced methods in hard OOD detection. ↓ (or ↑) indicates smaller (or larger) values are preferred; a bold font indicates the best results in a column.

| Methods | LSUN-Fix | | ImageNet-Resize | | CIFAR-100 | |
|---|---|---|---|---|---|---|
| | FPR95 ↓ | AUROC ↑ | FPR95 ↓ | AUROC ↑ | FPR95 ↓ | AUROC ↑ |
| KNN | 25.76 | 95.00 | 40.65 | 86.30 | 64.50 | 86.32 |
| OE | 10.45 | 98.33 | 14.95 | 97.78 | 53.55 | 90.40 |
| CSI | 34.85 | 91.72 | 33.30 | 90.50 | 45.64 | 87.64 |
| SSD+ | 23.95 | 95.74 | 53.52 | 84.00 | 46.87 | 90.50 |
| DOE | **3.00** | **99.15** | **7.20** | **98.55** | **41.55** | **91.85** |

(e.g., Mahalanobis and KNN). Furthermore, compared with the OE-based methods (i.e., OE and MixOE), other fine-tuning methods only show comparable, even inferior, performance in OOD detection. Therefore, the OE-based methods that utilize surrogate OOD remain hard to beat among state-of-the-art methods, even with its inherent OOD distribution gap issue.

Further, the DOE's improvement in OOD detection is notable compared to OE and MixOE, with 7.26 and 8.40 better results on the CIFAR-10 dataset, and with 20.30 and 26.66 better results on the CIFAR-100 dataset. Note that the tiny-ImageNet dataset is adopted as the surrogate OOD data, which is largely different from the considered test OOD datasets. Thus, we emphasize that the improvement of our method is due to our novel distributional-robust learning scheme, mitigating the OOD distribution gap between the surrogate and the unseen OOD cases.

We emphasize that the improvement of our DOE compared with that of OE is not dominated by our specific choice of scoring strategy. To verify this, we conduct experiments with OE and DOE and then employ the MaxLogit scoring after model training. Figure 2 illustrates the scoring densities with (a) OE and (b) DOE on the CIFAR-100 dataset, where we consider two test-time OOD datasets, namely, Texture and SVHN. Compared with that of OE, the overlap regions of DOE between the ID (i.e., CIFAR-10) and the OOD (i.e., Texture and SVHN) distributions are reduced. It reveals that even with the same

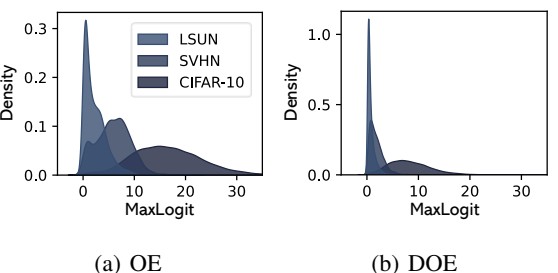

(a) OE        (b) DOE

Figure 2: The scoring densities of OE and DOE on CIFAR-100 dataset, where the MaxLogit is employed.

scoring function (i.e., MaxLogit), DOE can still improve the model's detection capability compared with the original OE. Therefore, we state that the key reason for our improved performance is our novel learning strategy, learning from extra OOD data that can benefit the model. Please refer to Appendix C for their detailed comparison.

**ImageNet Benchmark.** Huang & Li (2021) show that many advanced methods developed on the CIFAR benchmarks can hardly work for the ImageNet dataset due to its large semantic space with about 1k classes. Therefore, Table 1 also compares the results of DOE with advanced methods on ImageNet. As we can see, similar to the cases with CIFAR benchmarks, the fine-tuning approaches generally reveal superior results compared with the post-hoc approaches, and DOE remains effective in showing the best detection performance in expectation. Overall, Table 1 demonstrates the effectiveness of DOE across widely adopted experimental settings, revealing the power of our implicit data search scheme and distributional robust learning scheme.

## 5.2 HARD OOD DETECTION

Besides the above test OOD datasets, we also consider hard OOD scenarios (Tack et al., 2020), of which the test OOD data are very similar to that of the ID cases in style. Following the common setup (Sun et al., 2022) with the CIFAR-10 dataset being the ID case, we evaluate our DOE on three hard OOD datasets, namely, LSUN-Fix (Yu et al., 2015), ImageNet-Resize (Deng et al., 2009), and CIFAR-100. Note that data in ImageNet-Resize (1000 classes) with the same semantic space as

Table 3: Effectiveness of implicit data transformation and distributional robust learning. ↓ (or ↑) indicates smaller (or larger) values are preferred; a bold font indicates the best results in a row.

| | Implicit Data Transformation | | | Distributional Robust Learning | | | DOE | OE |
|---|---|---|---|---|---|---|---|---|
| | All-ones | Gaussian | Uniform | $\chi^2$ | WD | AT | | |
| FPR95 ↓ | 38.30 | 32.78 | 32.50 | 46.93 | 42.85 | 45.24 | **25.38** | 45.68 |
| AUROC ↑ | 92.67 | 91.25 | 91.55 | 89.17 | 90.51 | 90.45 | **93.97** | 87.61 |

tiny-ImageNet (200 classes) are removed. We compare our DOE with several works reported to perform well in hard OOD detection, including KNN, OE, CSI, and SSD+, where the results are summarized in Table 2. As we can see, our DOE can beat these advanced methods across all the considered datasets, even for the challenging CIFAR-10 versus CIFAR-100 setting. To some extent, it may indicate that our implicit data synthesis can even cover some hard OOD cases, and thus our DOE can lead to improved performance in hard OOD detection.

## 5.3 ABLATION STUDY

Our proposal claims two key contributions. The first one is the implicit data transformation via model perturbation, and the second one is the distributional robust learning scheme regarding WOR. Here, we design a series of experiments to demonstrate their respective power.

**Implicit Data Transformation.** In Section 3.1, we demonstrate that model perturbation can lead to data transformation. Here, we verify that other realizations (besides searching for WOR) can also benefit the model with additional OOD data. We employ perturbation with fixed values of ones (all-ones) and two types of random noise, namely, Gaussian noise with 0 mean and $I$ covariance matrix (Gaussian) and uniform noise over the interval $[-1, 1]$ (Uniform) (cf., Appendix B). We summarize their results on CIFAR-100 in Table 3 (Implicit Data Transformation). Compared to MSP and OE without model perturbation, all the forms of perturbation can lead to improved detection, indicating that our implicit data transformation is general to benefit the model with additional OOD data.

**Distributional Robust Learning.** In Section 4, we employ the implicit data transformation for uniform performance in OOD detection. As mentioned in Section 1, DRO (Rahimian & Mehrotra, 2019) also focuses on distributional robustness. Here, we conduct experiments with two realizations of DRO, with $\chi^2$ divergence ($\chi^2$) (Hashimoto et al., 2018) and Wasserstein distance (WD) (Kwon et al., 2020) (cf., Appendix B). We also consider the adversarial training (AT) (Madry et al., 2018b) as a baseline method, which can also be interpreted from the lens of DRO.

We summarize the related experiments on CIFAR-100 in Table 3 (Distributional Robust Learning). For two traditional DRO-based methods (i.e., $\chi^2$ and WD), they mainly consider the cases where the support of the test OOD data is a subset of the surrogate case. This close-world setup fails in OOD detection, and thus they reveal unsatisfactory results. Though AT also makes data transformation, its transformation is limited to additive noise, which can hardly cover the diversity of unseen data. In contrast, our DOE can search for complex transform functions that exploit unseen, having large improvements compared to all other robust learning methods.

## 6 CONCLUSION

Our proposal makes two key contributions. The first is the implicit data transformation for OOD synthesis, based on our novel insight that model perturbation leads to data transformation. Synthetic data follow a diverse distribution compared to original ones, rendering the target model to learn from unseen data. The second contribution is a distributional-robust learning method, building upon a min-max optimization scheme in searching for the worst regret. We demonstrate that learning from the worst regret in OOD detection can demonstrate better results than the risk-based counterpart. Accordingly, we propose DOE to mitigate the OOD distribution gap issue inherent in OE-based methods, where the extensive experiments verify our effectiveness. Our two contributions may not be limited to the OOD detection field. We will explore their usage scenarios in other areas, such as OOD generalization, adversarial training, and distributionally robust optimization.

## 7 ACKNOWLEDGMENTS

QZW and BH were supported by NSFC Young Scientists Fund No. 62006202, Guangdong Basic and Applied Basic Research Foundation No. 2022A1515011652, RGC Early Career Scheme No. 22200720, RGC Research Matching Grant Scheme No. RMGS20221102, No. RMGS20221306 and No. RMGS20221309. BH was also supported by CAAI-Huawei MindSpore Open Fund and HKBU CSD Departmental Incentive Grant. TLL was partially supported by Australian Research Council Projects IC-190100031, LP-220100527, DP-220102121, and FT-220100318.

## 8 ETHIC STATEMENT

This paper does not raise any ethical concerns. This study does not involve any human subjects, practices to data set releases, potentially harmful insights, methodologies and applications, potential conflicts of interest and sponsorship, discrimination/bias/fairness concerns, privacy and security issues, legal compliance, and research integrity issues.

## 9 REPRODUCIBILITY STATEMENT

The experimental setups for training and evaluation as well as the hyper-parameters are described in detail in Section 5, and the experiments are all conducted using public datasets. The code is publicly available at: github.com/qizhouwang/doe.

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

## A    PROOFS

This section provides the detailed proofs for our theoretical claims in the main text.

### A.1    PROOF OF PROPOSITION 1

*Proof.* To make the derivation clear, we adopt the equivalent form for our recursive definition of the model in equation 3, following:

$$h^{(l+1)}(W^{(l)}\mathbf{z}^{(l)}) = h^{(l+1)}(\mathbf{z}^{(l)}; W^{(l)}). \tag{11}$$

Then, by multiplicatively perturbing the $l$-th layer of the model, we have

$$
\begin{aligned}
h^{(l+1)}\left(\mathbf{z}^{(l)}; W^{(l)}(I + \alpha A^{(l)})\right) &= \max\left\{\left(W^{(l)}(I + \alpha A^{(l)})\right)\mathbf{z}^{(l)}, 0\right\} \\
&= \max\left\{W^{(l)}\left((I + \alpha A^{(l)})\mathbf{z}^{(l)}\right), 0\right\} \\
&= h^{(l+1)}\left((I + \alpha A^{(l)})\mathbf{z}^{(l)}; W^{(l)}\right).
\end{aligned}
\tag{12}
$$

Therefore, measuring in feature space $\mathcal{Z}^{(l)}$, multiplicative perturbation modifies the original features $\mathbf{z}^{(l)}$ by an affine transformation $I + \alpha A^{(l)}$. Assuming that the original data are i.i.d. drawn from the distribution with the probability density function (pdf) $f_{Z^{(l)}}(\mathbf{z}^{(l)})$, then the transformed data are i.i.d. drawn from the distribution with the pdf $f_{Z'^{(l)}}(\mathbf{z}'^{(l)}) = f_{Z^{(l)}}(\mathbf{z}^{(l)})\left|I + \alpha A^{(l)}\right|^{-1}$.

Using the KL-divergence to measure the discrepancy between the original feature distribution and the transformed feature distribution, we have

$$D_{\mathrm{KL}}(f_{Z^{(l)}} || f_{Z'^{(l)}}) = \mathbb{E}_{f_{Z^{(l)}}(\mathbf{z}^{(l)})} \log \frac{f_{Z^{(l)}}(\mathbf{z}^{(l)})}{f_{Z'^{(l)}}(\mathbf{z}'^{(l)})} = \log\left|I + \alpha A^{(l)}\right|. \tag{13}$$

Without loss of generality, we assume $K$ different eigenvalues for the matrix $A^{(l)}$. Then, by the Jordan matrix decomposition, we can write $A^{(l)} = T^{(l),-1}J^{(l)}T^{(l)}$. Therein, $J^{(l)}$ is of the form:

$$
\begin{bmatrix}
J(\lambda_1) & & & & & \\
& J(\lambda_2) & & & & \\
& & \ddots & & & \\
& & & J(\lambda_k) & & \\
& & & & \ddots & \\
& & & & & J(\lambda_K)
\end{bmatrix},
\tag{14}
$$

and $J(\lambda_k)$ is the $k$-th Jordan block (of size $n_k \times n_k$) corresponding to the $k$-th eigenvalue of the matrix $A^{(l)}$. Then, we have $\left|I + \alpha A^{(l)}\right| = \left|T^{(l),-1}(I + \alpha J^{(l)})T^{(l)}\right| = \left|I + \alpha J^{(l)}\right|$. Since $J^{(l)}$ is an upper triangular matrix, we can write $\left|I + \alpha J^{(l)}\right| = \prod_{k=1}^{K}(\alpha\lambda_k + 1)^{n_k}$. Accordingly, if the eigenvalues of the matrix $A^{(l)}$ are all greater than 0 and $\alpha > 0$, we have $\left|I + \alpha A^{(l)}\right| > 1$ and $D_{\mathrm{KL}}(f_{Z^{(l)}} || f_{Z'^{(l)}}) > 0$. Therefore, the distributions $f_{Z^{(l)}}$ and $f_{Z'^{(l)}}$ are different regarding the KL divergence. Thus we complete our proof. $\qquad\square$

### A.2    PROOF OF THEOREM 1

*Proof.* We consider an induction proof, justifying that: the multiplicative perturbation with $A^{(l)} \in \mathbb{R}^{n_l \times n_l}$ of any layer in $l = 1, \ldots, L$ can be transformed into an equivalent multiplicative perturbation with $\bar{A}^{(l-1)} \in \mathbb{R}^{n_{l-1} \times n_{l-1}}$ in the $(l-1)$-th layer. Moreover, $|\bar{A}^{(l-1)}| > 0$ if $|A^{(l)}| > 0$. Then, one can transform the multiplicative perturbation of the model to an equivalent form in the input space. Since the determinant of the equivalent perturbation is greater than 0, by applying Proposition 1, we conclude that multiplicative perturbation can lead to data transformation in the original input space.

To find the equivalent perturbation matrix $\bar{A}^{(l-1)}$ in the $(l-1)$-th layer regarding the original one $A^{(l)}$ in the $l$-th layer, we solve the following equation:

$$W^{(l)}(I + \alpha A^{(l)})h^{(l)}(W^{(l-1)}\mathbf{z}^{(l-1)}) = W^{(l)}h^{(l)}(W^{(l-1)}(I + \alpha\bar{A}^{(l-1)})\mathbf{z}^{(l-1)}). \tag{15}$$

If $[\boldsymbol{z}^{(l-1)}]_i \neq 0$ in each dimension, equation 15 can be rewritten as

$$A^{(l)}h^{(l)}(W^{(l-1)}\boldsymbol{z}^{(l-1)}) = h^{(l)'}(W^{(l-1)}\boldsymbol{z}^{(l-1)})W^{(l-1)}\bar{A}^{(l-1)}\boldsymbol{z}^{(l-1)}, \tag{16}$$

by applying the Taylor Theorem for the right-hand side[1]. Then, since the ReLU activation is applied, we solve the equivalent formulation for equation 16, following,

$$A^{(l)}W^{(l-1)} = W^{(l-1)}\bar{A}^{(l-1)}. \tag{17}$$

Then, the solution of $\bar{A}^{(l-1)}$ is $W^{(l-1),\dagger}A^{(l)}W^{(l-1)}$ with $\dagger$ being the Moore-Penrose inverse.

We justify that the multiplicative perturbation in the $l$-th layer can be transformed to that of the $(l-1)$-th layer. Therefore, the equivalent perturbation $\bar{A}^{(l-1)}$ and the original perturbation in the $(l-1)$-th layer can formulate a joint perturbation $\bar{\bar{A}}^{(l-1)}$, namely, $I + \alpha\bar{\bar{A}}^{(l-1)}$, with

$$\bar{\bar{A}}^{(l-1)} = \bar{A}^{(l-1)} + A^{(l-1)} + \alpha A^{(l-1)}\bar{A}^{(l-1)}. \tag{18}$$

Now, we justify that $\bar{\bar{A}}^{(l-1)}$ can also lead to distributional transformation. If $W^{(l-1),\dagger} = W^{(l-1),-1}$ (Here, we implicitly assume that $n_{l-1} = n_{l-2}$) and the eigenvalues of the matrix $A^{(l)}$ are all greater than 0, then we know that the eigenvalues of the matrix $\bar{A}^{(l-1)}$ are all greater than 0. Again, we have $\left|I + \alpha\bar{A}^{(l-1)}\right| > 1$. Then, the joint perturbation $\bar{\bar{A}}^{(l-1)}$ satisfies:

$$\left|I + \alpha\bar{\bar{A}}^{(l-1)}\right| = \left|(I + \alpha A^{(l-1)})(I + \alpha\bar{A}^{(l-1)})\right| \tag{19}$$

$$= \left|I + \alpha A^{(l-1)}\right|\left|I + \alpha\bar{A}^{(l-1)}\right| \tag{20}$$

$$> \left|I + \alpha A^{(l-1)}\right| \tag{21}$$

$$> 1. \tag{22}$$

By induction, the multiplicative perturbation of the model can be approximated by the input transformation. By applying Proposition 1, we know that $\boldsymbol{x}$ and the perturbation-based transformed counterpart follow the different data distributions. Thus we complete our proof. □

### A.3 Proof of Lemma 1

*Proof.* For the $L + 1$-layer ReLU network, we assume its model parameters and the model perturbation are the same as that of the corresponding layers for the $L$-layer ReLU network (except for the $L + 1$-th layer). Then, by inspecting equation 20, the perturbation from the $L + 1$-th layer can make the perturbation matrices for the $L + 1$-layer network no smaller than that of the $L$-layer network regarding each layer (including the input space) of the joint multiplicative perturbation. Thus, we complete our proof. □

### A.4 Excess Risk Bound

We further derive the learning bound of DOE. Here, we make the standard assumptions for our learning problem. First, we assume that the Rademacher Complexity $\mathfrak{R}_n(\mathcal{H})$ of $\mathcal{H}$ is bounded, i.e., there is a $C_{\mathcal{H}}$ such that $\mathfrak{R}_n(\mathcal{H}) \leq C_{\mathcal{H}}/\sqrt{n}$, holding for ReLU models. Further, the CE loss is bounded by $A_{\text{CE}}$ and is $L_{\text{CE}}$ Lipschitz continuous; the OE loss is bounded by $A_{\text{OE}}$ and is $L_{\text{OE}}$ Lipschitz continuous. To ease notation, we also define

$$\epsilon(C, L, A) = 2CL + A\sqrt{\frac{\log 1/\delta}{2}}. \tag{23}$$

We are now ready to state the upper bound for the worst-case population performance of our DOE.

**Theorem 2.** *Given ID and surrogate OOD training sample $S_{ID}$ and $S_{OOD}$, we write the optimal solution as $h_W^* = \arg\min_{h_W \in \mathcal{H}} \mathcal{L}_{DOE}(h_W; D_{ID}, D_{OOD}^s)$ and the empirical counterpart as $\hat{h}_W = \arg\min_{h_W \in \mathcal{H}} \mathcal{L}_{DOE}(h_W; S_{ID}, S_{OOD}^s)$. Then, under above assumptions, w.p. at least $1 - \delta$, we have*

$$\mathcal{L}_{DOE}(\hat{h}_W; D_{ID}, D_{OOD}^s) \leq \mathcal{L}_{DOE}(h_W^*; D_{ID}, D_{OOD}^s)$$

$$+ (2 + 4\lambda)\epsilon(C_{\mathcal{H}}, L, A)/\sqrt{\min\{|S_{ID}|, |S_{OOD}^s|\}}, \tag{24}$$

*where $L = \max\{L_{CE}, L_{OE}\}$ and $A = \max\{A_{CE}, A_{OE}\}$.*

---

[1]With the usual adjustments that the equations only hold almost everywhere in parameter space.

*Proof.* We apply the Rademacher Bound for $\mathcal{L}_{\mathrm{CE}}$ and $\mathcal{L}_{\mathrm{OE}}$, given that w.p. at least $1 - \sigma$, we have

$$|\mathcal{L}_{\mathrm{CE}}(h_{\mathrm{W}}; D_{\mathrm{ID}}) - \mathcal{L}_{\mathrm{CE}}(h_{\mathrm{W}}; S_{\mathrm{ID}})| \leq \epsilon(C_{\mathcal{H}}, L_{\mathrm{CE}}, A_{\mathrm{CE}})/\sqrt{|S_{\mathrm{ID}}|}, \tag{25}$$

$$|\mathcal{L}_{\mathrm{OE}}(h_{\mathrm{W}}; D_{\mathrm{OOD}}) - \mathcal{L}_{\mathrm{OE}}(h_{\mathrm{W}}; S_{\mathrm{OOD}})| \leq \epsilon(C_{\mathcal{H}}, L_{\mathrm{OE}}, A_{\mathrm{OE}})/\sqrt{|S_{\mathrm{OOD}}|}, \tag{26}$$

for all $h_{\mathrm{W}} \in \mathcal{H}$. When the hypothesis space $\mathcal{H}$ is large enough, we have

$$h_{\mathrm{W}}^* = \arg\min_{h_{\mathrm{w}} \in \mathcal{H}} \mathcal{L}_{\mathrm{CE}}(h_W; D_{\mathrm{ID}}) = \arg\min_{h_{\mathrm{w}} \in \mathcal{H}} \max_{P:||P|| \leq \rho} \mathrm{Regret}_{\mathrm{OE}}(h_{W+\alpha P}; D_{\mathrm{OOD}}^{\mathrm{s}}). \tag{27}$$

Accordingly, by the definition of $\mathcal{L}_{\mathrm{CE}}(h_{\mathrm{W}}^*; D_{\mathrm{ID}})$, for any $\epsilon > 0$, there exists $h_{\mathrm{W}}^\epsilon$ such that $\mathcal{L}_{\mathrm{CE}}(h_{\mathrm{W}}^\epsilon; D_{\mathrm{ID}}) \leq \mathcal{L}_{\mathrm{CE}}(h_{\mathrm{W}}^*; D_{\mathrm{ID}}) + \epsilon$. Thus, using $\mathcal{L}_{\mathrm{CE}}(\hat{h}_{\mathrm{W}}; S_{\mathrm{ID}}) \leq \mathcal{L}_{\mathrm{CE}}(h_{\mathrm{W}}^\epsilon; S_{\mathrm{ID}})$, we can write

$$\mathcal{L}_{\mathrm{CE}}(\hat{h}_{\mathrm{W}}; D_{\mathrm{ID}}) - \mathcal{L}_{\mathrm{CE}}(h_{\mathrm{W}}^*; D_{\mathrm{ID}})$$

$$= \mathcal{L}_{\mathrm{CE}}(\hat{h}_{\mathrm{W}}; D_{\mathrm{ID}}) - \mathcal{L}_{\mathrm{CE}}(h_{\mathrm{W}}^\epsilon; S_{\mathrm{ID}}) + \mathcal{L}_{\mathrm{CE}}(h_{\mathrm{W}}^\epsilon; S_{\mathrm{ID}}) - \mathcal{L}_{\mathrm{CE}}(h_{\mathrm{W}}^*; D_{\mathrm{ID}}) \tag{28}$$

$$\leq \mathcal{L}_{\mathrm{CE}}(\hat{h}_{\mathrm{W}}; D_{\mathrm{ID}}) - \mathcal{L}_{\mathrm{CE}}(h_{\mathrm{W}}^\epsilon; S_{\mathrm{ID}}) + \epsilon \tag{29}$$

$$= \mathcal{L}_{\mathrm{CE}}(\hat{h}_{\mathrm{W}}; D_{\mathrm{ID}}) - \mathcal{L}_{\mathrm{CE}}(\hat{h}_{\mathrm{W}}; S_{\mathrm{ID}}) + \mathcal{L}_{\mathrm{CE}}(\hat{h}_{\mathrm{W}}; S_{\mathrm{ID}}) - \mathcal{L}_{\mathrm{CE}}(h_{\mathrm{W}}^\epsilon; D_{\mathrm{ID}}) + \epsilon \tag{30}$$

$$\leq \mathcal{L}_{\mathrm{CE}}(\hat{h}_{\mathrm{W}}; D_{\mathrm{ID}}) - \mathcal{L}_{\mathrm{CE}}(\hat{h}_{\mathrm{W}}; S_{\mathrm{ID}}) + \mathcal{L}_{\mathrm{CE}}(h_{\mathrm{W}}^\epsilon; S_{\mathrm{ID}}) - \mathcal{L}_{\mathrm{CE}}(h_{\mathrm{W}}^\epsilon; D_{\mathrm{ID}}) + \epsilon \tag{31}$$

$$\leq 2 \sup_{h \in \mathcal{H}} |\mathcal{L}_{\mathrm{CE}}(h; D_{\mathrm{ID}}) - \mathcal{L}_{\mathrm{CE}}(h; S_{\mathrm{ID}})| + \epsilon. \tag{32}$$

Since equation 25 holds for $h_{\mathrm{W}} \in \mathcal{H}$ and $\epsilon > 0$, we have

$$\mathcal{L}_{\mathrm{CE}}(\hat{h}_{\mathrm{W}}; D_{\mathrm{ID}}) \leq \mathcal{L}_{\mathrm{CE}}(h_{\mathrm{W}}^*; D_{\mathrm{ID}}) + 2\epsilon(C_{\mathcal{H}}, L_{\mathrm{CE}}, A_{\mathrm{CE}})/\sqrt{|S_{\mathrm{ID}}|}. \tag{33}$$

For any $h_{\mathrm{W}} \in \mathcal{H}$, we also have

$$\sup_{P:||P|| \leq \rho} \mathrm{Regret}_{\mathrm{OE}}^{\alpha,P}(\hat{h}_W; D_{\mathrm{OOD}}^{\mathrm{s}})$$

$$\leq \sup_{P:||P|| \leq \rho} \left[ \mathcal{L}_{\mathrm{OE}}(\hat{h}_{W+\alpha P}; S_{\mathrm{OOD}}) - \min_{W^*} \mathcal{L}_{\mathrm{OE}}(h_{W^*+\alpha P}; S_{\mathrm{OOD}}) \right] + \frac{2\epsilon(C_{\mathcal{H}}, L_{\mathrm{OE}}, A_{\mathrm{OE}})}{\sqrt{|S_{\mathrm{OOD}}|}} \tag{34}$$

$$\leq \sup_{P:||P|| \leq \rho} \left[ \mathcal{L}_{\mathrm{OE}}(\hat{h}_{W+\alpha P}; S_{\mathrm{OOD}}) - \min_{W^*} \mathcal{L}_{\mathrm{OE}}(h_{W^*+\alpha P}; S_{\mathrm{OOD}}) \right] + \frac{2\epsilon(C_{\mathcal{H}}, L_{\mathrm{OE}}, A_{\mathrm{OE}})}{\sqrt{|S_{\mathrm{OOD}}|}} \tag{35}$$

$$\leq \sup_{P:||P|| \leq \rho} \left[ \mathcal{L}_{\mathrm{OE}}(h_{W+\alpha P}; S_{\mathrm{OOD}}) - \min_{W^*} \mathcal{L}_{\mathrm{OE}}(h_{W^*+\alpha P}; S_{\mathrm{OOD}}) \right] + \frac{2\epsilon(C_{\mathcal{H}}, L_{\mathrm{OE}}, A_{\mathrm{OE}})}{\sqrt{|S_{\mathrm{OOD}}|}} \tag{36}$$

$$\leq \sup_{P:||P|| \leq \rho} \left[ \mathcal{L}_{\mathrm{OE}}(h_{W+\alpha P}; D_{\mathrm{OOD}}) - \min_{W^*} \mathcal{L}_{\mathrm{OE}}(h_{W^*+\alpha P}; D_{\mathrm{OOD}}) \right] + \frac{4\epsilon(C_{\mathcal{H}}, L_{\mathrm{OE}}, A_{\mathrm{OE}})}{\sqrt{|S_{\mathrm{OOD}}|}} \tag{37}$$

$$\leq \sup_{P:||P|| \leq \rho} \mathrm{Regret}_{\mathrm{OE}}^{\alpha,P}(h_W; D_{\mathrm{OOD}}^{\mathrm{s}}) + \frac{4\epsilon(C_{\mathcal{H}}, L_{\mathrm{OE}}, A_{\mathrm{OE}})}{\sqrt{|S_{\mathrm{OOD}}|}}, \tag{38}$$

indicating that

$$\sup_{P:||P|| \leq \rho} \mathrm{Regret}_{\mathrm{OE}}^{\alpha,P}(\hat{h}_W; D_{\mathrm{OOD}}^{\mathrm{s}}) \leq \sup_{P:||P|| \leq \rho} \mathrm{Regret}_{\mathrm{OE}}^{\alpha,P}(h_W^*; D_{\mathrm{OOD}}^{\mathrm{s}}) + \frac{4\epsilon(C_{\mathcal{H}}, L_{\mathrm{OE}}, A_{\mathrm{OE}})}{\sqrt{|S_{\mathrm{OOD}}|}}. \tag{39}$$

Combining equation 33 and equation 39, we complete our proof. $\square$

The theorem states that the empirical solution leads to a promising detection capability in expectation, which considers the uniform OOD performance via the WOR. The critical point is that the original surrogate OOD is still very important (i.e., the small sample size of $S_{\mathrm{OOD}}$ leads to loose excess bound), even if our method can synthesize additional OOD data.

---

**Algorithm 1** Distribution-agnostic Outlier Exposure (DOE).

**Input:** ID and OOD samples from $D_{\text{ID}}$ and $D_{\text{OOD}}^{\text{s}}$, resp;
$P_{\text{MA}} = 0$;
**for** $\texttt{ns} = 1$ **to** $\texttt{num\_step}$ **do**
    Sample $B_{\text{ID}}$ and $B_{\text{OOD}}^{\text{s}}$ from ID and surrogate OOD, resp;
    $P = 0$;
    **if** $\texttt{ns} > \texttt{num\_warm}$ **then**
        **for** $\texttt{np} = 1$ **to** $\texttt{num\_pert}$ **do**
            $\text{WOR}_{\text{G}}(h_W; B_{\text{OOD}}^{\text{s}}) = ||\nabla_{\sigma|\sigma=1.0}\mathcal{L}_{\text{OE}}(\sigma \cdot h_{W+\alpha P}; B_{\text{OOD}}^{\text{s}})||^2$;
            $P \leftarrow \nabla_P \text{WOR}_{\text{G}}(h_{W+\alpha P}; B_{\text{OOD}}^{\text{s}})$;
        **end for**
        $P_{\text{MA}} \leftarrow (1-\beta) \cdot P_{\text{MA}} + \beta \cdot \text{NORM}(P)$;
        $W \leftarrow W - \texttt{lr} \cdot \nabla_W [\mathcal{L}_{\text{CE}}(h_W; B_{\text{ID}}) + \lambda \mathcal{L}_{\text{OE}}(h_{W+\alpha P_{\text{MA}}}; B_{\text{OOD}}^{\text{s}})]$;
    **else**
        $W \leftarrow W - \texttt{lr} \cdot \nabla_W [\mathcal{L}_{\text{CE}}(h_W; B_{\text{ID}}) + \lambda \mathcal{L}_{\text{OE}}(h_W; B_{\text{OOD}}^{\text{s}})]$;
    **end if**
**end for**
**Output:** detection model $h_W(\cdot)$.

---

# B   Algorithm Designs

We summarize details of algorithm designs for a set of related learning schemes.

## B.1   Distributional Robustness and Distribution Gap

Overall, to demonstrate why our distributional-robust learning scheme can mitigate the OOD distribution gap, we consider the following two situations: (1) the true OOD distribution contains all the different OOD situations; and (2) the capacity of implicit data transformation is large enough.

For the first situation, we assume that the true OOD distribution contains all the different OOD situations, i.e., all samples with labels out of the considered label space. It is a reasonable consideration since we do not know what kinds of OOD data will be encountered during the test, and thus all the different OOD situations can be encountered. In this case, the surrogate and the (associated) implicit OOD data are subsets of the true OOD distribution since they do not have overlapped semantics with the ID distribution. Then, compared with OE that learns only from surrogate OOD data, our DOE can further benefit from implicit OOD data. It can enlarge the coverage of OOD situations since implicit data follows new data distributions over the surrogate OOD distribution (cf., Theorem 1).

For the second situation, we assume that the capacity of implicit data transformation is large enough to cover sufficiently many OOD cases. This is also a reasonable assumption since the transformation's capacity can benefit from layer-wise architectures (cf., Lemma 1), and deep models (which contain many layers) are typically adopted in OOD detection. Accordingly, although we do not know precisely what is the true OOD distribution, we can upper-bound the worst OOD performance to guarantee uniform performance of the model under various test situations (cf., Theorem 2). When the capacity is large enough (covering many test OOD situations), DOE performs well under these unseen test OOD data, thus mitigating the OOD distribution gap.

## B.2   DOE

Algorithm 1 summarizes a stochastic realization of our DOE. The overall algorithm is run for $\texttt{num\_step}$ steps, with $\texttt{num\_warm}$ epochs of warm-up in employing the original OE. Then, in each training step, we first calculate the perturbation $P$ regarding the OOD mini-batch for $\texttt{num\_pert}$ steps, and the normalized results are used to update the moving average $P_{\text{MA}}$. With the resultant perturbation $P_{\text{MA}}$ for the OE loss, we update the model via one step of mini-batch gradient descent. After training, we apply the MaxLogit scoring in discerning ID and OOD cases.

### B.3 WORST RISK-BASED DOE

Our proposed DOE searches for the model perturbation that leads to WOR, which is the worst regret-based realization. In our main text, we state its superior to the risk-based counterpart in Section 4, with the experimental verification in Section 5.3. For integrity, we further describe the realization of the worst risk-based DOE, named DOE-risk.

Similar to our proposed DOE, DOE-risk can also be formalized by a min-max learning problem:

$$\mathcal{L}_{\text{DOE}}(h_W; D_{\text{ID}}, D_{\text{OOD}}^{\text{s}}) = \mathcal{L}_{\text{CE}}(h_W; D_{\text{ID}}) + \lambda \max_{P:||P|| \leq 1} \mathcal{L}_{\text{OE}}(h_{W+\alpha P}; D_{\text{OOD}}^{\text{s}}). \tag{40}$$

Then, for its stochastic realization, one step of gradient ascent is employed for the perturbation with respect to the mini-batch, namely,

$$P \leftarrow \nabla_{P|P=0} \mathcal{L}_{\text{OE}}(h_{W+\alpha P}; B_{\text{OOD}}^{\text{s}}). \tag{41}$$

All other parts follow the realization of the original DOE. After training, we also employ the MaxLogit scoring in discerning ID and OOD data.

### B.4 IMPROVED OE WITH PREDEFINED PERTURBATION

We consider several implicit data transformations with the predefined perturbations in Section 5.3. Here, we briefly summarize their realizations.

**All-ones Matrices.** For the perturbation matrices with fixed values, we employ the simple all-ones matrices, namely, $P_{\text{one}} = \{I^{(l)}\}_{l=1}^L$, with $I^{(l)} \in \mathbb{R}^{n_{l-1} \times n_{l-1}}$ for $l = 1, \ldots, L$ being the all-ones matrix. Then, the associated learning objective can be written as:

$$\mathcal{L}_{\text{OE-one}}(h_W; D_{\text{ID}}, D_{\text{OOD}}^{\text{s}}) = \mathcal{L}_{\text{CE}}(h_W; D_{\text{ID}}) + \lambda \mathcal{L}_{\text{OE}}(h_{W+\alpha P_{\text{one}}}; D_{\text{OOD}}^{\text{s}}). \tag{42}$$

**Gaussian Noise.** When adopting Gaussian noise for random perturbation, we have $P_{\text{gau}} = \{N^{(l)}\}_{l=1}^L$, with the elements drawn from Gaussian distribution with $0$ mean and $1$ standard deviation. Then, the associated learning objective is of the form

$$\mathcal{L}_{\text{OE-gau}}(h_W; D_{\text{ID}}, D_{\text{OOD}}^{\text{s}}) = \mathcal{L}_{\text{CE}}(h_W; D_{\text{ID}}) + \lambda \mathcal{L}_{\text{OE}}(h_{W+\alpha P_{\text{gau}}}; D_{\text{OOD}}^{\text{s}}). \tag{43}$$

**Uniform Noise.** Similarly, one can adopt uniform noise for random perturbation, which we denote by $P_{\text{uni}} = \{U^{(l)}\}_{l=1}^L$. The elements of $U^{(l)}$ are drawn from the uniform noise over the interval $[-1, 1]$. Then, the associated learning objective is

$$\mathcal{L}_{\text{OE-uni}}(h_W; D_{\text{ID}}, D_{\text{OOD}}^{\text{s}}) = \mathcal{L}_{\text{CE}}(h_W; D_{\text{ID}}) + \lambda \mathcal{L}_{\text{OE}}(h_{W+\alpha P_{\text{uni}}}; D_{\text{OOD}}^{\text{s}}). \tag{44}$$

### B.5 DRO

The *distirbutionally robust optimization* (DRO) (Rahimian & Mehrotra, 2019) is a traditional technique to make the model perform uniformly well. In OE, one can utilize DRO by replacing the original OE risk in equation 2 with its distributional robust counterpart, namely,

$$\mathcal{L}^{\text{DRO}}(h; D_{\text{ID}}, D_{\text{OOD}}^{\text{s}}) = \mathcal{L}_{\text{CE}}(h; D_{\text{ID}}) + \lambda \underbrace{\sup_{D_{\text{OOD}}^{\text{w}} \in \mathcal{U}(D_{\text{OOD}}^{\text{s}})} \mathcal{L}_{\text{OE}}(h; D_{\text{OOD}}^{\text{w}})}_{\mathcal{L}_{\text{OE}}^{\text{DRO}}(h; D_{\text{OOD}}^{\text{s}})}, \tag{45}$$

where $\mathcal{U}(D_{\text{OOD}}^{\text{s}})$ is the *ambiguity set*. Basically, $\mathcal{U}(D_{\text{OOD}}^{\text{s}})$ constrains the difference between the surrogate OOD distribution $D_{\text{OOD}}^{\text{s}}$ and its worst counterpart $D_{\text{OOD}}^{\text{w}}$. In expectation, equation 45 makes the training procedure cover a wide range of potential test OOD distributions in $\mathcal{U}(D_{\text{OOD}}^{\text{s}})$, guaranteeing its uniform performance by bounding the worst OOD risk derived by $D_{\text{OOD}}^{\text{w}}$.

The ambiguity set $\mathcal{U}(D_{\text{OOD}}^{\text{s}})$ is defined by $\{D^{\text{w}} : \text{Div}_f(D^{\text{w}}||D) \leq \rho\}$ with $\text{Div}_f(\cdot)$ the $f$-divergence and $\rho$ the constraint. For the worst OOD distribution that leads to the worst OOD risk, a weighting-based searching scheme can be derived for the empirical counterpart of equation 45, following the form of re-weighted empirical risk, namely,

$$\sup_{\boldsymbol{p}} \sum p\ell_{\text{OE}}(h(\boldsymbol{x})) \text{ s.t. } \boldsymbol{p} \in \{\boldsymbol{p} \mid \boldsymbol{p} \in \Delta \text{ and } D_f(\boldsymbol{p}||\mathbf{1}) \leq \rho\}. \tag{46}$$

However, due to this equivalent re-weighting scheme, DRO actually assumes that the support of test-time OOD data is among that of the training situation. This assumption is violated in OOD detection since the surrogate OOD data can be largely different from the unseen situations, i.e., their support sets can be greatly different. Therefore, traditional DRO cannot lead to much improved results compared with original OE, which we demonstrate by the experimental results in Section 5.3. Note that some DRO methods (Krueger et al., 2021) try to search for worst distributions that go beyond the support set of training data. However, they rely on more than one training domain, which is not directly applicable in OOD detection.

$\chi^2$**-divergence DRO.** Hashimoto et al. (2018) define the ambiguity set by the $\chi^2$ divergence, given by $D_{\chi^2}(P||Q) = \int \left(\frac{dP}{dQ} - 1\right)^2 dQ$. They assume that data distribution can be written as the joint form of the sub-populations, i.e., $D_{\mathrm{OOD}}^{\mathrm{s}} = \sum_{k \in [K]} \alpha_k D_{\mathrm{OOD}}^{\mathrm{s,k}}$. Then, one can derive the dual form of the $\mathcal{L}_{\mathrm{OE}}^{\mathrm{DRO}}(h; D_{\mathrm{OOD}}^{\mathrm{s}})$ in equation 46 with respect to the $\chi^2$ divergence, namely,

$$\inf_{\eta \in \mathbb{R}} \left\{ (2(1/\alpha_{\min} - 1)^2 + 1)^{1/2} \left( \mathbb{E}_{D_{\mathrm{OOD}}^{\mathrm{s}}} \left[ \max \left\{ \ell_{\mathrm{OE}}(h(x)) - \eta, 0 \right\}^2 \right] \right)^{1/2} + \eta \right\}, \qquad (47)$$

where $\alpha_{\min} = \min_k \alpha_k$. Then, Hashimoto et al. suggest that for deep models that rely on stochastic gradient descent, one can utilize the dual objective in equation 47, leading to the learning objective of the form:

$$\mathcal{L}^{\chi^2}(h; D_{\mathrm{ID}}, D_{\mathrm{OOD}}^{\mathrm{s}}) = \mathcal{L}_{\mathrm{CE}}(h; D_{\mathrm{ID}}) + \lambda \mathbb{E}_{D_{\mathrm{OOD}}^{\mathrm{s}}} \left[ \max \left\{ \ell_{\mathrm{OE}}(h(x)) - \eta, 0 \right\}^2 \right], \qquad (48)$$

where $\eta$ is treated as a hyperparameter. Overall, equation 48 ignores all data points that suffer less than $\eta$-levels of loss values, while large loss above $\eta$ are upweighted due to the square operation.

**Wasserstein DRO.** The ambiguity set with the Wasserstein distance has also attracted much attention in the literature. Specifically, Wasserstein distance is given by

$$\mathcal{W}_r(P, Q) = \left( \inf_{O \in J(P,Q)} \left\{ \int_{\mathcal{Z} \times \mathcal{Z}} ||\zeta - \tilde{\zeta}||^r dO(\zeta, \tilde{\zeta}) \right\} \right)^{1/r}. \qquad (49)$$

However, the direct calculation for the Wasserstein DRO is intractable, and Kwon et al. (2020) propose a simple learning method that leads to its effective approximation. Specifically, if the loss function is differentiable and its gradient is Holder continuous, one can optimize the following surrogate objective as an effective approximation for the optimal solution of Wasserstein DRO:

$$\mathcal{L}^{\mathrm{WDRO}}(h; D_{\mathrm{ID}}, D_{\mathrm{OOD}}^{\mathrm{s}}) = \mathcal{L}_{\mathrm{CE}}(h; D_{\mathrm{ID}}) + \lambda \left( \mathcal{L}_{\mathrm{OE}}(h; D_{\mathrm{OOD}}) + \mathbb{E}_{D_{\mathrm{OOD}}} ||\nabla_{\boldsymbol{x}} \ell_{\mathrm{OE}}(h(\boldsymbol{x}))|| \right). \qquad (50)$$

Please refer to (Kwon et al., 2020) for an in-depth discussion.

## B.6 AT

Adversarial training (AT) (Madry et al., 2018a) directly modifies input features that lead to increased risk, which can also be interpreted from the lens of distributional robust learning (Sinha et al., 2018). For OE, one can modify features for surrogate OOD data by adding adversarial noise, namely,

$$\delta_{\mathrm{p}} \leftarrow \mathtt{Proj} \left[ \delta_{\mathrm{p}} + \kappa \mathtt{sign} \left( \nabla_{\delta_{\mathrm{p}}} \ell_{\mathrm{OE}}(h(x + \delta_{\mathrm{p}})) \right) \right], \qquad (51)$$

where $\kappa$ controls the magnitude of the perturbation, $\mathtt{Proj}$ is the clipping operation for the valid $\delta_{\mathrm{p}}$, and $\mathtt{sign}$ is the signum function. Equation 51 iterates for several steps and $\delta_{\mathrm{p}}$ is typically initialized by random noise.

Applying the adversarial noise for surrogate OOD data, the resultant learning objective is

$$\mathcal{L}^{\mathrm{AT}}(h; D_{\mathrm{ID}}, D_{\mathrm{OOD}}^{\mathrm{s}}) = \mathcal{L}_{\mathrm{CE}}(h; D_{\mathrm{ID}}) + \lambda \mathbb{E}_{D_{\mathrm{OOD}}^{\mathrm{s}}} \left[ \ell_{\mathrm{OE}}(h(x + \delta_{\mathrm{p}})) \right]. \qquad (52)$$

AT can be viewed as a direct way of data transformation. However, as demonstrated in Section 5.3, the associated transformation function is simpler than our DOE. Therefore, the performance of AT is inferior to our DOE.

Table 4: Comparison of DOE and advanced methods on CIFAR-10 dataset. ↓ (or ↑) indicates smaller (or larger) values are preferred; a shaded row of results indicate the best method in previous post-hoc (or fine-tuning) methods; and a bold font indicates the best result in a column.

| Method | SVHN | | LSUN | | iSUN | | Texture | | Places365 | | Average | | ID ACC |
|---|---|---|---|---|---|---|---|---|---|---|---|---|---|
| | FPR95↓ | AUROC↑ | FPR95↓ | AUROC↑ | FPR95↓ | AUROC↑ | FPR95↓ | AUROC↑ | FPR95↓ | AUROC↑ | FPR95↓ | AUROC↑ | |
| Post-hoc Approach | | | | | | | | | | | | | |
| MSP | 65.60 | 81.23 | 23.05 | 96.74 | 56.55 | 89.59 | 61.45 | 87.47 | 62.20 | 86.95 | 53.77 | 88.40 | 94.28 |
| ODIN | 55.30 | 83.65 | 8.55 | 98.48 | 36.95 | 92.66 | 54.00 | 84.34 | 59.20 | 84.33 | 42.80 | 88.69 | 94.28 |
| Mahalanobis | 9.35 | 98.00 | 45.15 | 92.90 | 37.15 | 93.54 | 11.80 | 97.92 | 71.45 | 83.68 | 34.98 | 93.21 | 94.28 |
| Free Energy | 55.40 | 76.92 | 3.65 | 99.21 | 31.65 | 93.26 | 52.80 | 84.20 | 45.35 | 87.77 | 37.77 | 88.27 | 94.28 |
| ReAct | 76.95 | 74.91 | 0.95 | 99.21 | 64.15 | 83.19 | 81.00 | 72.59 | 68.05 | 81.16 | 58.22 | 82.21 | 94.28 |
| KNN | 31.29 | 95.01 | 26.84 | 95.33 | 29.48 | 94.28 | 41.21 | 92.08 | 44.02 | 90.47 | 34.56 | 93.43 | 94.28 |
| Fine-tuning Approach | | | | | | | | | | | | | |
| OE | 4.30 | 99.12 | 0.85 | 99.76 | 11.45 | 98.17 | 17.35 | 97.03 | 28.10 | 95.17 | 12.41 | 97.85 | 94.58 |
| CSI | 20.48 | 96.63 | 6.18 | 98.78 | 5.49 | 98.99 | 21.07 | 96.27 | 33.73 | 93.68 | 17.39 | 96.87 | 94.33 |
| SSD+ | 0.28 | 99.10 | 4.07 | 98.71 | 35.96 | 95.28 | 8.90 | 98.34 | 25.00 | 95.40 | 14.84 | 97.36 | 95.46 |
| MixOE | 22.35 | 96.21 | 1.25 | 99.68 | 7.75 | 98.65 | 17.15 | 96.75 | 19.25 | 96.68 | 13.55 | 97.59 | 94.79 |
| VOS | 35.20 | 91.57 | 6.15 | 98.85 | 26.95 | 93.65 | 49.35 | 85.06 | 40.10 | 88.69 | 31.55 | 91.56 | 95.45 |
| DOE | 2.65 | 99.36 | 0.00 | 99.89 | 0.75 | 99.67 | 7.25 | 98.47 | 15.10 | 96.53 | 5.15 | 98.78 | 94.18 |

Table 5: Comparison of DOE and advanced methods on the CIFAR-100 dataset. ↓ (or ↑) indicates smaller (or larger) values are preferred; a shaded row of results indicate the best method in post-hoc (or fine-tuning) methods; and a bold font indicates the best results in the a column.

| Method | SVHN | | LSUN | | iSUN | | Texture | | Places365 | | Average | | ID ACC |
|---|---|---|---|---|---|---|---|---|---|---|---|---|---|
| | FPR95↓ | AUROC↑ | FPR95↓ | AUROC↑ | FPR95↓ | AUROC↑ | FPR95↓ | AUROC↑ | FPR95↓ | AUROC↑ | FPR95↓ | AUROC↑ | |
| Post-hoc Approach | | | | | | | | | | | | | |
| MSP | 80.90 | 75.19 | 51.25 | 87.93 | 85.35 | 73.48 | 83.40 | 71.94 | 82.75 | 72.66 | 76.73 | 76.24 | 73.98 |
| ODIN | 70.75 | 72.57 | 63.38 | 76.55 | 60.23 | 74.83 | 60.31 | 76.96 | 61.61 | 77.72 | 63.25 | 75.72 | 73.98 |
| Mahalanobis | 58.45 | 86.54 | 99.80 | 52.40 | 34.70 | 92.96 | 44.40 | 90.13 | 80.50 | 68.14 | 65.57 | 78.03 | 73.98 |
| Free Energy | 89.70 | 73.09 | 16.90 | 96.96 | 86.45 | 75.20 | 82.75 | 73.76 | 82.00 | 73.56 | 71.56 | 78.51 | 73.98 |
| ReAct | 79.10 | 81.73 | 8.50 | 98.46 | 92.55 | 85.30 | 85.30 | 73.58 | 84.25 | 72.47 | 69.94 | 78.21 | 73.98 |
| KNN | 49.73 | 88.06 | 31.94 | 93.81 | 37.11 | 91.86 | 48.30 | 87.96 | 84.16 | 71.96 | 50.24 | 86.73 | 73.98 |
| Fine-tuning Approach | | | | | | | | | | | | | |
| OE | 55.15 | 85.62 | 11.65 | 96.92 | 48.40 | 87.90 | 47.35 | 87.02 | 65.85 | 80.57 | 45.68 | 87.61 | 75.33 |
| CSI | 62.96 | 84.75 | 96.47 | 49.28 | 95.91 | 52.98 | 78.30 | 71.25 | 85.00 | 71.45 | 83.72 | 65.94 | 74.30 |
| SSD+ | 13.30 | 97.45 | 82.55 | 86.03 | 38.74 | 91.69 | 71.24 | 82.52 | 77.41 | 79.20 | 56.65 | 87.38 | 75.91 |
| MixOE | 83.80 | 74.26 | 20.10 | 96.26 | 54.20 | 87.24 | 55.80 | 86.13 | 46.30 | 88.39 | 52.04 | 86.46 | 75.81 |
| VOS | 60.22 | 88.57 | 85.45 | 83.62 | 50.57 | 88.80 | 80.65 | 74.22 | 90.30 | 64.73 | 73.43 | 79.98 | 73.55 |
| DOE | 19.20 | 96.43 | 4.15 | 99.02 | 12.80 | 97.65 | 32.75 | 91.88 | 58.00 | 84.87 | 25.38 | 93.97 | 74.51 |

## C FURTHER EXPERIMENTS

This section provides further experiments to demonstrate the effectiveness of our proposal.

### C.1 CIFAR BENCHMARKS

We first summarize the main experiments in Table 4-5 on CIFAR benchmarks for the common OOD detection. A brief version can also be found in Table 1 in the main text. Overall, our DOE reveals superior performance on average regarding both the evaluation metrics of FPR95 and AUROC. However, when it comes to individual test-time OOD datasets, our DOE may not work best in all situations (e.g., KNN on OOD dataset Places365 and ID dataset CIFAR-100). We emphasize that it does not challenge the generality of our proposal since DOE has demonstrated stable improvements for the original OE. Here, the interesting point is that if we can further benefit the OE from the latest progress in OOD scoring, one can further improve the performance of our method in effective OOD detection, which requires our future study.

Now, we compare OE and DOE on CIFAR benchmarks with five individual trails in Table 6, where we report the mean results and the standard deviation. As we can see, our DOE not only leads to improved average performance in OOD detection, and the results are more stable than that of the original OE. The superiority of DOE in stability may lie in the fact that the target model can learn from more data than the OE case, further demonstrating the effectiveness of our proposal.

Also, we compare the performance of OE and DOE when using the MSP scoring and the MaxLogit scoring, of which the experiments are summarized in Table 7. Regarding both the cases with different scoring functions, our DOE always achieve superior performance than that of the OE, demonstrating that our proposal can genuinely mitigate the OOD distribution gap issue in OOD detection. Further, comparing the results across different scoring strategies, we observe that using the

Table 6: Comparison of DOE and OE on CIFAR benchmarks with 5 individual trails. ↓ (or ↑) indicates smaller (or larger) values are preferred; and a bold font indicates the best results in the corresponding column.

| Method | SVHN | | LSUN | | iSUN | | Texture | | Places365 | |
|---|---|---|---|---|---|---|---|---|---|---|
| | FPR95↓ | AUROC↑ | FPR95↓ | AUROC↑ | FPR95↓ | AUROC↑ | FPR95↓ | AUROC↑ | FPR95↓ | AUROC↑ |
| CIFAR-10 | | | | | | | | | | |
| OE | 2.91± 0.60 | 99.34± 0.08 | 0.46± 0.13 | 99.80± 0.03 | 9.05± 1.56 | 98.45± 0.20 | 18.08± 0.88 | 96.91± 0.11 | 28.02± 0.68 | 95.01± 0.08 |
| DOE | **2.66±** **0.10** | **99.41±** **0.01** | **0.15±** **0.01** | **99.91±** **0.01** | **1.26±** **0.11** | **99.48±** **0.04** | **7.40±** **0.27** | **98.32±** **0.02** | **15.42±** **0.42** | **96.33±** **0.03** |
| CIFAR-100 | | | | | | | | | | |
| OE | 55.48± 1.46 | 86.99± 0.99 | 12.28± 0.95 | 86.76± 0.19 | 44.38± 2.75 | 88.54± 0.90 | 47.57± 1.42 | 86.93± 0.21 | 65.05± 1.27 | 80.83± 0.23 |
| DOE | **28.47±** **0.35** | **95.45±** **0.02** | **5.27±** **0.18** | **98.51±** **0.01** | **22.28±** **0.92** | **96.36±** **0.10** | **40.00±** **0.54** | **91.34±** **0.06** | **50.70±** **0.27** | **88.42±** **0.03** |

Table 7: Comparison of DOE and OE on CIFAR and ImageNet benchmarks with the MSP scoring and the MaxLogit scoring. ↓ (or ↑) indicates smaller (or larger) values are preferred; and a bold font indicates the best results in the corresponding column.

| Method | CIFAR-10 | | | | CIFAR-100 | | | | ImageNet | | | |
|---|---|---|---|---|---|---|---|---|---|---|---|---|
| | MaxLogit | | MSP | | MaxLogit | | MSP | | MaxLogit | | MSP | |
| | FPR95↓ | AUROC↑ | FPR95↓ | AUROC↑ | FPR95↓ | AUROC↑ | FPR95↓ | AUROC↑ | FPR95↓ | AUROC↑ | FPR95↓ | AUROC↑ |
| OE | 11.07 | 97.98 | 12.41 | 97.85 | 35.95 | 92.42 | 45.68 | 87.61 | 71.25 | 79.18 | 73.80 | 78.90 |
| DOE | **5.15** | **98.78** | **7.83** | **98.46** | **25.38** | **93.97** | **30.50** | **92.75** | **59.83** | **83.54** | **65.20** | **80.83** |

MaxLogit scoring leads to better results than using the MSP scoring. Therefore, we choose the MaxLogit scoring in our DOE.

## C.2 IMAGENET BENCHMARKS

Table 8 lists the detailed experiments on the ImageNet benchmark. Overall, our DOE achieves superior performance on average against all the considered baselines. Further, for the cases with iNaturalist and Places365, which are believed to be the challenging OOD datasets on the ImageNet situation, our DOE also achieve considerable improvements against all other advanced methods. It demonstrates that our DOE can also work well for challenging detection scenarios with extremely large semantic space and complex data patterns.

## C.3 TRAINING FROM SCRATCH WITH DOE

This section further considers the training setup of training from scratch, where we mainly focus on the detection performance of OE and DOE on CIFAR benchmarks. Specifically, the models are trained for 150 epochs for OE and DOE via stochastic gradient descent. We fix the learning rate to be 0.1, divided by 10 per 30 epochs. For DOE, the warmup epochs are set to be 90. All other hyperparameters follow the same setup as in Section 5. We summarize the experimental results in Table 9. As we can see, our DOE can still improve OE by a large margin, revealing that our method is general in applying for the setup of training from scratch.

Table 9: Comparison of OE and DOE when training from scratch on CIFAR benchmarks.

| from scratch | CIFAR-10 | | CIFAR-100 | |
|---|---|---|---|---|
| | FPR95 | AUROC | FPR95 | AUROC |
| OE | 15.46 | 95.77 | 46.02 | 88.14 |
| DOE | **5.85** | **98.52** | **26.47** | **93.16** |

## C.4 WORST OOD REGRET AND WORST OOD RISK

Another issue related to distributional robustness is our definition of the worst OOD distribution. It is defined by the OOD regret $\mathcal{L}_{\text{OE}}(h; D) - \inf_{h' \in \mathcal{H}} \mathcal{L}_{\text{OE}}(h'; D)$, where we claim its superiority than its risk counterpart, i.e., $\mathcal{L}_{\text{OE}}(h; D)$ (cf., Appendix B).

Table 8: Comparison of DOE and advanced methods on ImageNet dataset. ↓ (or ↑) indicates smaller (or larger) values are preferred; a shaded row of results indicate the best method in post-hoc (or fine-tuning) methods; and a bold font indicates the best results in the corresponding column.

| Method | iNaturalist | | SUN | | Places365 | | Texture | | Average | | ID ACC |
|---|---|---|---|---|---|---|---|---|---|---|---|
| | FPR95 ↓ | AUROC ↑ | FPR95 ↓ | AUROC ↑ | FPR95 ↓ | AUROC ↑ | FPR95 ↓ | AUROC ↑ | FPR95 ↓ | AUROC ↑ | |
| Post-hoc Approach | | | | | | | | | | | |
| MSP | 72.98 | 77.22 | 80.89 | 74.24 | 76.69 | 77.81 | 70.73 | 78.58 | 75.32 | 76.96 | 74.55 |
| ODIN | 63.85 | 77.78 | 89.98 | 61.80 | 88.00 | 67.17 | 67.87 | 77.40 | 77.43 | 71.04 | 74.55 |
| Mahalanobis | 95.90 | 60.56 | 95.42 | 45.33 | 98.90 | 44.65 | 55.80 | 84.60 | 86.50 | 58.78 | 74.55 |
| Free Energy | 69.10 | 77.39 | 82.36 | 76.08 | 76.15 | 80.23 | 56.97 | 84.32 | 71.14 | 79.50 | 74.55 |
| ReAct | 56.11 | 84.94 | 82.79 | 75.87 | 75.00 | 80.72 | 70.37 | 82.16 | 70.31 | 81.42 | 74.55 |
| KNN | 65.40 | 83.73 | 75.62 | 77.33 | 79.20 | 74.34 | 40.80 | 86.45 | 64.75 | 80.91 | 74.55 |
| Fine-tuning Approach | | | | | | | | | | | |
| OE | 78.31 | 75.23 | 80.10 | 76.55 | 70.41 | 81.78 | 66.38 | 82.04 | 73.80 | 78.90 | 75.51 |
| CSI | 75.85 | 82.63 | 90.62 | 47.83 | 94.90 | 44.62 | 85.85 | 87.11 | 86.80 | 65.54 | 74.27 |
| SSD+ | 59.60 | 85.54 | 75.62 | 73.80 | 83.60 | 68.11 | 39.40 | 82.40 | 64.55 | 77.46 | **78.80** |
| MixOE | 80.51 | 74.30 | **74.62** | **79.81** | 84.33 | 69.20 | 58.00 | 85.83 | 74.36 | 77.28 | 74.62 |
| VOS | 94.83 | 57.69 | 98.72 | 38.50 | 87.75 | 65.65 | 70.20 | 83.62 | 87.87 | 61.36 | 74.43 |
| DOE | **55.87** | **85.98** | 80.94 | 76.26 | **67.84** | **83.05** | **34.67** | **88.90** | **59.83** | **83.54** | 75.50 |

Table 10 summarizes the results on the CIFAR-100 dataset in comparison between searching for the worst OOD regret (DOE-regret) and the worst OOD risk (DOE-risk). Therein, both realizations can improve results compared with the original OE. However, the regret-based DOE can reveal better results than the risk-based one, with 4.59 further improvement in FPR95. Here, the worst regret can better indicate the worst OOD distribution than the risk counterpart, and thus the DOE-regret, as employed in Algorithm 1, demonstrates superior results in Table 10.

Table 10: Roubst learning with worst OOD regret and worst OOD risk.

| Methods | DOE-regret | DOE-risk |
|---|---|---|
| FPR95 ↓ | **25.74** | 30.33 |
| AUROC ↑ | **94.25** | 94.01 |

### C.5    EFFECT OF HYPER-PARAMETERS

We study the effect of hyper-parameters on the final performance of our DOE, where we consider the trade-off parameter $\lambda$, the perturbation strength $\alpha$, the smoothing strength $\beta$, the perturbation steps `num_pert`, and the warm-up epochs `num_warm`. We also study the case of sub-model perturbation, where the model perturbation is only applied to a part of the whole model. All the above experiments are conducted on the CIFAR-100 dataset.

As one can see from the Tables 11- 14, our DOE is pretty robust to different choices of the hyper-parameters (i.e., $\lambda$, $\beta$, `num_pert`, and `num_warm`), and the results are superior to the OE across most of the hyper-parameter settings. However, a proper choice of the hyper-parameters can truly induce improved results in effective OOD detection, reflecting that all the introduced hyper-parameters are useful in our proposed DOE. In Table 15, we further demonstrate that randomly selected $\alpha$ (from the candidates) reveals superior performance than assigning fixed values. Note that random selection can cover a wider range of OOD situations than that of fixed values. Then, since the model can learn from more implicit OOD data, the capability of the model in OOD detection is better than the case with fixed values.

Finally, we show the experimental results with sub-model perturbation in Table 16, where only a part of the model is perturbed in our DOE. Here, we separate the WRN-40-2 into 3 blocks, following the block structure in (Zagoruyko & Komodakis, 2016). As we can see, perturbing the whole model can reveal superior performance than the cases with sub-model perturbation. It can be explained by our Lemma 1 in that perturbing the whole model can benefit the data transformation from the layer-wise structure of deep models most. Then, with the more flexible form of transform function, perturbing the whole model can reveal better results than the cases with sub-model perturbation since the model can learn from more diverse (implicit) OOD data.

Table 11: DOE on CIFAR-100 with various $\lambda$.

|  | FPR95 | AUROC |
|---|---|---|
| 0.1 | 41.43 | 90.82 |
| 0.5 | 32.35 | 93.66 |
| 1.0 | 25.80 | 93.77 |
| 1.5 | 27.59 | 94.20 |
| 2.0 | **25.52** | **94.35** |
| 2.5 | 25.84 | 94.34 |
| 3.0 | 26.04 | 94.31 |
| 3.5 | 25.60 | 94.32 |
| 4.0 | 25.96 | 94.28 |
| 4.5 | 26.35 | 94.23 |

Table 12: DOE on CIFAR-100 with various $\beta$.

|  | FPR95 | AUROC |
|---|---|---|
| 0.1 | 23.52 | 94.48 |
| 0.2 | 24.31 | 64.36 |
| 0.3 | **23.12** | **94.47** |
| 0.4 | 23.90 | 94.41 |
| 0.5 | 24.79 | 94.25 |
| 0.6 | 25.11 | 94.22 |
| 0.7 | 26.37 | 93.95 |
| 0.8 | 28.12 | 93.71 |
| 0.9 | 30.23 | 93.19 |
| 1.0 | 27.25 | 93.77 |

Table 13: DOE on CIFAR-100 with various num_pert.

|  | FPR95 | AUROC |
|---|---|---|
| 1 | 25.59 | 94.50 |
| 2 | 24.90 | 94.83 |
| 3 | 26.75 | 94.22 |
| 4 | 25.37 | 94.41 |
| 5 | 24.62 | 94.83 |
| 6 | 25.37 | 94.40 |
| 7 | 24.60 | 94.38 |
| 8 | 25.80 | 94.15 |
| 9 | 24.83 | 94.71 |
| 10 | **24.54** | **94.88** |

Table 14: DOE on CIFAR-100 with various num_warm.

|  | FPR95 | AUROC |
|---|---|---|
| 1 | 25.50 | 94.09 |
| 2 | 25.21 | 93.94 |
| 3 | 24.81 | 94.03 |
| 4 | **23.96** | **94.39** |
| 5 | 25.50 | 94.11 |
| 6 | 26.18 | 93.79 |
| 7 | 26.02 | 94.49 |
| 8 | 29.01 | 94.07 |
| 9 | 36.05 | 92.19 |
| 10 | 35.33 | 92.89 |

Table 15: DOE on CIFAR-100 with various $\alpha$.

| candidate $\alpha$ | FPR95 | AUROC |
|---|---|---|
| $\{1e^{-1}\}$ | 53.20 | 86.49 |
| $\{1e^{-2}\}$ | 27.29 | 94.04 |
| $\{1e^{-3}\}$ | 31.75 | 93.71 |
| $\{1e^{-4}\}$ | 32.72 | 93.63 |
| $\{1e^{-1}, 1e^{-2}\}$ | 36.14 | 91.21 |
| $\{1e^{-2}, 1e^{-3}\}$ | 28.56 | 94.17 |
| $\{1e^{-3}, 1e^{-4}\}$ | 34.35 | 93.49 |
| $\{1e^{-1}, 1e^{-2}, 1e^{-3}\}$ | 27.59 | 94.20 |
| $\{1e^{-2}, 1e^{-3}, 1e^{-4}\}$ | 30.83 | 93.69 |
| $\{1e^{-1}, 1e^{-2}, 1e^{-3}, 1e^{-4}\}$ | **25.20** | **94.33** |

Table 16: DOE on CIFAR-100 with sub-model perturbation.

| Perturbed Block | FPR95 | AUROC |
|---|---|---|
| Block 1 | 35.76 | 93.82 |
| Block 2 | 32.88 | 93.73 |
| Block 3 | 32.50 | 93.54 |
| Block 1-2 | 27.29 | 94.06 |
| Block 2-3 | 31.89 | 93.63 |
| Block 1 and Block 3 | 28.07 | 94.04 |
| Whole Model | **25.25** | **94.47** |

