# OpenReview forum: "Out-of-distribution Detection with Implicit Outlier Transformation"
_ICLR.cc/2023/Conference — ICLR 2023 poster_

### Official Review · Reviewer_r8Mp · 2022-10-23

**Confidence:** 4
**Correctness:** 3
**Technical Novelty And Significance:** 3
**Empirical Novelty And Significance:** Not applicable
**Recommendation:** 6

**Clarity, Quality, Novelty And Reproducibility:**

Clarity: Generally clear, with some questions remaining (see above).

Quality: Generally high-quality, but with a lot of grammatical issues (which I believe are fixable with any grammar-correcting software, or a proof-reader).

Novelty: To my knowledge, using parameter-perturbation to synthesize surrogate outliers is a novel idea, and the overall algorithm also has smaller elements of novelty in it.

Reproducibility: There appears to be sufficient details to reproduce the main algorithm.

**Strength And Weaknesses:**

The key idea in the submission, that of using parameter-perturbation to simulate data-perturbation in the context of outlier-synthesis, is novel (to my knowledge). The results indicate effectiveness and potential superiority over alternatives. The method seems well-designed and implemented overall.

There are a few things that aren’t clear to me, and I hope the authors can clarify these:

  1. From Algorithm 1 and the description in Section 5, it appears that the method is strictly applied in a fine-tuning sense, i.e. once a network is trained with the classification loss, further fine-tuning training is performed with the DOE loss. It is unclear to me if the OE, MixOE, VOS baselines are also trained similarly. In the original papers, they do not appear to require a fine-tuning setup (except when pre-training with ImageNet-classification is performed, which does not seem the case for CIFAR in this submission).

  2. It is not obvious to me how the regret formulation connects precisely to Eq. 7. My understanding is that the gradient-norm approximation is used whenever the regret-formulation is referred to in experiments. Without a clear relation between WOR and the gradient-norm variant, Table 4 is hard to interpret: if DOE-regret doesn’t actually clearly correspond to the gradient-norm approximation, the comparison between gradient-norm and DOE-risk might suggest something else.

  3. In Appendix A.1, in Eq. 13, it looks like the inequality det(A+B) >= det(A) + det(B) is being used. This is generally not true, and only holds with conditions upon A and B. The statement of Minkowski’s inequality suggests that this is true when A and B are both positive semi-definite. (Potential typos: (a) After Eq. 14, it says v, which should probably be z. (b) In A.3, some of the (L+1)s should probably be L.)

Other comments:

  — Figure 1 contrasts DRO and DOE as performing interpolation and interpolation+extrapolation. REx (Krueger et al., 2021) shows how DRO might be modified to change constraints for coefficients lying in [0, 1] to lying anywhere, but summing to 1, which enable an extrapolation-variant for DRO.

  — The writing has plenty of grammatical issues, which can be fixed by getting the draft proof-read, or using a grammar-fixing software (there are free alternatives available).

**Summary Of The Paper:**

The submission proposes a potential improvement over the popular and effective outlier exposure (OE) method of Hendrycks et al. (2019). While existing works have explored improving the diversity of surrogate outliers in the input space (by synthesizing artificial examples/applying MixUp/resampling) and in the feature space (by perturbing feature representations of surrogate outliers), the proposed method involves perturbing the weights of the neural network instead to generate features corresponding to a new synthesized data distribution of surrogate outliers. The perturbation at each iteration of training is computed by gradient ascent to maximize a gradient-norm penalty, inspired from Arjovsky et al. (2019).

Experiments suggest that the proposed method outperforms other OE-based approaches.

**Summary Of The Review:**

While the theoretical discussion involves significant departure from practical implementation (many assumptions that are going to be un-met in practice), and there is some lack of clarity about how fairly the method is empirically compared to baselines, the general idea is intuitive, well-executed, and appears to work well enough that my overall recommendation at this time is to Accept.

---

> ### Author Response · Authors · 2022-11-14
> **Response to Reviewer r8Mp**
>
> We sincerely thank you for your constructive comments! Please find our responses below.
>
> > Q1. It is unclear to me if the OE, MixOE, VOS baselines are also trained similarly. In the original papers, they do not appear to require a fine-tuning setup.
>
> A1. Thank you for the concern. We double-check the source codes of these methods, confirming that OE and MixOE involve the fine-tuning setup while VOS is trained from scratch.
>
> In fact, DOE can still lead to the improved results over OE for the setup of training from scratch. The results on CIFAR benchmarks are listed in the following table. As we can see, for DOE, our default fine-tuning strategy can lead to better results than training from scratch. However, even for DOE training from scratch, its improvement over OE is also obvious. So, we conclude the effectiveness of our method is not limited to the learning scheme of fine-tuning. We will add a related discussion in our revision.
>
> | FPR95 | CIFAR-10 | CIFAR-100 |
> |-------|----------|-----------|
> |  OE   |   12.41  |  45.68    |
> |  DOE  |   **5.15**   | **25.38**     |
> |  DOE (from scratch)  | 6.97 | 27.68 |
>
> > Q2. It is not obvious to me how the regret formulation connects precisely to Eq. 7.
>
> A2. Intuitively, if the gradient norm is large, the current model is far from the optimum, and thus the corresponding regret should be large. Therefore, by minimizing/maximizing the gradient norm, one can accordingly minimize/maximize the regret. A similar trick is discussed in [1] (Sec. 1.1 Related Work), relating the gradient norm and the regret estimation.
>
> [1] Alekh Agarwal and Tong Zhang. "Minimax Regret Optimization for Robust Machine Learning under Distribution Shift", in *COLT*, 2022.
>
> > Q3. In Appendix A.1, in Eq. 13, it looks like the inequality det(A+B) >= det(A) + det(B) is being used. This is generally not true, and only holds with conditions upon A and B.
>
> A3. Thank you for the correction. In our revision, the assumption about determinants of perturbation matrices is **changed to a new assumption** about eigenvalues of perturbation matrices, i.e., all eigenvalues are greater than 1.  By applying the Jordan-Chevalley decomposition, we see that $\log \lvert I+\alpha A^{(l)}\rvert>0$. The related discussions and proofs in Appendix A.2 (for Theorem 1) and A.3 (for Lemma 1) are also modified accordingly.
>
>
>
> > Q4. Figure 1 contrasts DRO and DOE as performing interpolation and interpolation+extrapolation. REx (Krueger et al., 2021) shows how DRO might be modified to change constraints for coefficients lying in [0, 1] to lying anywhere, but summing to 1, which enable an extrapolation-variant for DRO.
>
> A4. Thank you for the suggestion. Unfortunately, REx is not directly applicable for the common setup in OOD detection, due to their reliance on more than one training domain. Therefore, it is not discussed in our paper. However, it is a very interesting method that can motivate future research in OOD detection, involving a novel setup where we have more than one training domain (with respect to OOD cases). We will add a related discussion in our revision.
>
> > Q5. The writing has plenty of grammatical issues.
>
> A5. Sorry for the writing typos. We will fix the grammar issues in our revision.

---

> > ### Comment · Reviewer_r8Mp · 2022-11-22
> > **Follow-up**
> >
> > Thanks for the response!
> >
> > Q1. Thanks for confirming fine-tuning in the two baselines, and investigating the improvements when trained from scratch.
> >
> > Q2. The connection between gradient norm and WOR is still not very clear to me. Agarwal and Zhang describe that one can view IRM as a “minimax formulation similar to the [regret formulation]”, but there seems to be no clear connection  between a regret objective and the IRMv1 instantiation, as far as I can tell. The proposed intuition seems reasonable though.
> >
> > Q3. It is not clear to me how the Jordan-Chevalley decomposition relates to the inequality being proved. From my understanding, the Jordan-Chevalley decomposition tells us that a square matrix may be expressed as a sum of a diagonalizable matrix and a nilpotent matrix, with additional properties satisfied by these matrices (they commute, and are polynomials in the original matrix). Can the authors describe further why this decomposition implies the inequality, with the assumption that the eigen-values of A are positive? What are the implications for the perturbation matrix needing to have positive eigenvalues? Does this not preclude a large class of perturbations?

---

> > > ### Author Response · Authors · 2022-11-23
> > > **Thank you for the further feedback.**
> > >
> > > Sincerely thanks for your further feedback. Please find our responses below.
> > >
> > > > Q2. The connection between gradient norm and WOR is still not very clear to me. Agarwal and Zhang describe that one can view IRM as a “minimax formulation similar to the [regret formulation]”, but there seems to be no clear connection between a regret objective and the IRMv1 instantiation, as far as I can tell. The proposed intuition seems reasonable though.
> > >
> > > A2. In [1], one can interpret IRM as a min-max regret formulation of the form $f_\text{IRM}=\arg\inf_f R(f), \text{s.t.}, \sup_P R_P(f) - R^*_P\le\epsilon$, where the constraint condition links the regret and the gradient norm computation in IRM. It motivates our method in using the gradient norm for WOR estimation, of which the intuition is discussed in Section 4.2. Also, we are glad to hear that you would like to support our intuition.
> > >
> > > Overall, our method of using gradient norm for WOR estimation is relatively new, but it is actually motivated by the discussion in [1]. Therefore, we cite [1] in our paper to appreciate its exciting  contribution.
> > >
> > > [1] Alekh Agarwal and Tong Zhang. "Minimax Regret Optimization for Robust Machine Learning under Distribution Shift", in COLT, 2022.
> > >
> > > > Q3. It is not clear to me how the Jordan-Chevalley decomposition relates to the inequality being proved.
> > >
> > > A3. Sorry for the confusing discussion. We use the Jordan matrix decomposition (instead of Jordan-Chevalley Decomposition), following $A=T^{-1}J T$, where $J$ is the upper triangle matrix with diagonal elements corresponds to $n$ eigenvalues of $A$ (written as $\lambda_i$ for $i=1,\ldots, n$ in the following). Since the determinant of an upper triangular matrix is the multiplication of its diagonal elements, we have $\lvert I+\alpha A \rvert=\lvert T^{-1}(I+\alpha J) T \rvert=\lvert I+\alpha J \rvert=\prod_i (1+\alpha \lambda_i)$ (note that $I+\alpha J$ is still an upper triangular matrix). Accordingly, when  $\lambda_i>0$ for $i=1,\ldots,n$, we have $\prod_i (1+\alpha \lambda_i)>1$ and $\lvert I+\alpha A \rvert>1$.
> > >
> > > We will updated the related discussion in our paper. Thanks again for your constructive feedback.

---

> ### Author Response · Authors · 2022-11-18
> **We are looking forward to your responses or further suggestions/comments!**
>
> Dear Reviewer r8Mp,
>
> We have addressed your initial concerns regarding our paper. We are happy to discuss them with you in the openreview system if you still have some concerns/questions. We also welcome new suggestions/comments from you!
>
> Best regards,
>
> Authors of #363

---

> ### Author Response · Authors · 2022-11-22
> **We are looking forward to your responses or further suggestions/comments!**
>
> Dear Reviewer r8Mp,
>
> We appreciate your efforts and time in providing constructive feedback and comments. Could you check our response and confirm whether we have solved your concern? If there are further questions or suggestions, we will address them to improve our submission.
>
> Sincerely,
>
> Authors of #363

---

### Official Review · Reviewer_tVEd · 2022-10-25

**Confidence:** 5
**Correctness:** 2
**Technical Novelty And Significance:** 2
**Empirical Novelty And Significance:** Not applicable
**Recommendation:** 5

**Clarity, Quality, Novelty And Reproducibility:**

The paper is well written. The originality is good. The reproducibility seems to be negative due to the huge amount of parameter searching efforts and training (pre-training) tricks required.

**Strength And Weaknesses:**

The strengths of the work include:
- The key idea of implicit outlier transformation is interesting. Synthesizing model perturbation and the outlier exposure to create a set of outliers of different distributions is new. It is similar to a combination of deep ensemble and OE, but it has an explicit searching for model parameters more suitable for the OOD detection task.
- Theoretical results are given to support that the resulting transformed outliers has a different distribution from its original distribution.
- The method is extensively evaluated against both fine-tuning and post-hoc approaches on commonly used benchmarks.
- The paper is well written and easy to follow.

However, the paper also has a number of limitations:
- The proposed method significantly complicate the OE method. The effective performance of the method relies heavily on a large set of parameter searching and training (pre-training) tricks. This largely limits its real-life applications.
- Compared to OE, the proposed method uses a different OOD scoring function, MaxLogit vs MSP in OE. As mentioned in the paper, MaxLogit is much better than MSP, so it is not clear the actual improvement of the proposed implicit OE method gained over the OE method. The paper shows a small set of results of comparing OE and the proposed method using the same scoring function MaxLogit and MSP, but the experiments are limited to only a few experiment cases there. It would be more straightforward by directly using MaxLogit in the OE baseline on all the datasets. This concern substantially weakens my confidence on the proposed method.
- The presented theoretical results have some points there, but they are not that relevant in the sense that although the method can produce a different distribution from the original distribution, it is not clear whether the resulting distribution is closer to the true distribution of OOD data than the original distribution. in other words, both of the original distribution and the transformed distribution play a similar role wrt ID data; they are both surrogate OOD data only: OOD distribution vs. its shifted version.
- The statement "It means that for target data of our interest, we can implicitly get them access via finding the corresponding model perturbation. Then, by updating the detection model thereafter, it can learn from the transformed data (i.e., the target ones) instead of original inputs." is an unjustified over-claim. As discussed in the above point, the transformed data is still a set of outliers, some shifted version of the original outliers, which are not the target OOD data in the test data.
- I wonder whether this would affect the classification accuracy. The empirical results show that the method does not. It would be good to elaborate the reasons in the paper.

**Summary Of The Paper:**

The paper introduces an outlier exposure(OE)-enhanced method for OOD detection, which searches for model perturbation options to transform a set of given outlier data to its transformed version. It shows through experiments on commonly used benchmarks that this approach is better than OE and a number of other similar approaches.

**Summary Of The Review:**

The overall idea is interesting, but its theoretical and empirical support have some major issues. Some key statements are unjustified. Overall, the paper may benefit from another round of major revision.

---

> ### Author Response · Authors · 2022-11-14
> **Response to Reviewer tVEd [2/2]**
>
> > Q5. I wonder whether this would affect the classification accuracy. The empirical results show that the method does not. It would be good to elaborate the reasons in the paper.
>
> A5. Thank you for your valuable comment. In general, deep models have the sufficient capacity to learn from diverse OOD data. Since our DOE is no more than introducing additional OOD data for training, it is reasonable to observe that the classification accuracy is not largely affected.
>
> Similar phenomena is also observed in recent works, such as VOS [1], which also regularizes the model with additional OOD data. We have discussed the difference between our method and VOS in the Introduction, and the experimental results demonstrated the superiority of our method over VOS in Section 5.
>
>
> [1] Xuefeng Du, et al. "VOS: Learning What You Don't Know By Virtual Outlier Synthesis", in *ICLR* 2022.
>
>
> > Q6. The reproducibility seems to be negative due to the huge amount of parameter searching efforts and training (pre-training) tricks required.
>
> A6. We have the validation OOD dataset that is different from the test OOD data, and thus it is reasonable to do hyper-parameter searching. Furthermore, as demonstrated in A1 and Appendix C.3 (Tables 10~13), our DOE is pretty robust to different choices of hyper-parameters, in that most hyper-parameter settings (except for some extreme choices of the values) lead to the improved performance over OE. Further, we release the code by the [anonymous Github repository](https://anonymous.4open.science/r/DOE/README.md). We will add a formal link if the paper is accepted.

---

> ### Author Response · Authors · 2022-11-14
> **Response to Reviewer tVEd [1/2]**
>
> We sincerely thank you for your constructive comments! Please find our responses below.
>
> > Q1. The proposed method significantly complicates the OE method. The effective performance of the method relies heavily on a large set of parameter searching and training (pre-training) tricks. This largely limits its real-life applications.
>
> A1. Thank you for your valuable comment. First, we have the validation set for hyper-parameter tuning, which is seperated from the surrogate OOD data (different from the test situation). Therefore, it is valid for our DOE to conduct parameter searching. We will emphasize the experimental setup in our revision.
>
> Also, we have conducted the hyper-parameter analysis in Appendix C.3 (Tables 10~13), demonstrating that our DOE is better than traditional OE over most of different choices of hyper-parameters. For the further verification, the following table lists the results on CIFAR benchmarks, where we fix $\lambda=1$, $\beta=1$ (without moving average), and $warm up=0$ (without pre-training). As we can see, our DOE still reveals promising results over the baselines (such as OE and MSP).
>
> | FPR95 | CIFAR-10 | CIFAR-100 |
> |-------|----------|-----------|
> |  MSP  | 53.77    | 76.73     |
> |  OE   | 12.41    | 45.68     |
> |  DOE  | **7.45**     | **27.40**     |
>
>
>
> > Q2. It would be more straightforward by directly using MaxLogit in the OE baseline on all the datasets. This concern substantially weakens my confidence on the proposed method.
>
> A2. Thank you for your suggestion. In our paper, we have conducted experiments on CIFAR benchmarks in Appendix C.3 (Table 8), where we compare DOE with OE under MaxLogit scoring. The results verify the superiority of DOE over OE, with respect to MaxLogit scoring.
>
>
> As your suggestion, we further conduct experiments with MaxLogit scoring on ImageNet. The results are listed in the following table, further demonstrating that the implicit data transformation and the distributional-agnostic learning scheme mainly attribute to the improved results of DOE.
>
> |  ImageNet    | FPR95    | AUROC     |
> |--------------|----------|-----------|
> |  OE-MaxLogit | 71.25    | 79.18     |
> |  DOE         | **59.83**    | **83.54**     |
>
> > Q3. It is not clear whether the resulting distribution is closer to the true distribution of OOD data than the original distribution.
>
> A3. Thank you for the concern. We consider the following two situations where our DOE can mitigate the OOD distribution gap: (1) *the true OOD distribution contains all the different OOD situations*; and (2) *the capacity of implicit data transformation is large enough*.
>
> **For the first situation**, we assume that the true OOD distribution contains all the different OOD situations, i.e., all those data with labels out of the considered label space. It is a reasonable consideration, since we do not know what kinds of OOD data will be encountered during the test, and thus all the different OOD situations should be considered. In this case, the surrogate and the (associated) implicit OOD data are subsets of the true OOD distribution, since they do not have overlapped semantics with the ID distribution. Then, compared with OE that learns only from surrogate OOD data, our DOE can further benefit from implicit OOD data. It can enlarge the coverage of OOD situations, because implicit data follows new data distributions over the surrogate OOD distribution (cf., Theorem 1).
>
> **For the second situation**, we assume that the capacity of implicit data transformation is large enough, in that it can cover sufficiently many OOD cases. This is also a reasonable assumption since the capacity of the transformation can benefit from layer-wise architectures (cf., Lemma 1 and Table 15 in Appendix C.3), and deep models (which contain many layers) are typically adopted in OOD detection. Accordingly, although we do not know precisely what is the true OOD distribution, we can upper-bound the worst OOD performance to guarantee the uniform performance of the model under various test situations (cf., Theorem 2). When the capacity is large enough (covering many test OOD situations), DOE performs well under these unseen test OOD data, thus mitigating the OOD distribution gap.
>
> The above justification will be added to our revision.
>
> > Q4. The transformed data is still a set of outliers, some shifted version of the original outliers, which are not the target OOD data in the test data.
>
> A4. Sorry for the confusing point. As mentioned in A3, the model is learned from the worst OOD distribution to ensure the uniform performance of the model under various test OOD situations. Here, we refer the target OOD data as the **worst OOD data** that can benefit the model most (instead of the real OOD data), which is learnable by maximizing the OOD regret (cf., Definition 2). We will make it more precise in our revision.

---

> ### Author Response · Authors · 2022-11-18
> **We are looking forward to your responses or further suggestions/comments!**
>
> Dear Reviewer tVEd,
>
> We have addressed your initial concerns regarding our paper. We are happy to discuss them with you in the openreview system if you still have some concerns/questions. We also welcome new suggestions/comments from you!
>
> Best regards,
>
> Authors of #363

---

> ### Author Response · Authors · 2022-11-22
> **We are looking forward to your responses or further suggestions/comments!**
>
> Dear Reviewer tVEd,
>
> We appreciate your efforts and time in providing constructive feedback and comments. Could you check our response and confirm whether we have solved your concern? If there are further questions or suggestions, we will address them to improve our submission.
>
> Sincerely,
>
> Authors of #363

---

### Official Review · Reviewer_UUPv · 2022-10-25

**Confidence:** 4
**Correctness:** 4
**Technical Novelty And Significance:** 4
**Empirical Novelty And Significance:** 4
**Recommendation:** 8

**Clarity, Quality, Novelty And Reproducibility:**

The paper is well written and easy to follow. However, with no code available it is untrivial to implement the proposed methods.
Hyper-parameters are not discussed on how they were chosen.

**Strength And Weaknesses:**

Strengths:
1) A solution tackling the distribution gap between used auxiliary ood data and that encountered in practice.
2)The method is supported with theoretical evidence.
3) Good results.
Weakness:
1) Some closely related works are missing e.g., [1], [2] & [3].
2) Since the proposed perturbation are parameters based, it is not well discussed to which layers are the perturbations applied or the role of the used architecture. Not sure why Wide ResNet is used for Cifar experiments while ResNet 50 for ImageNet.




[1] Chen, Jiefeng, et al. "Atom: Robustifying out-of-distribution detection using outlier mining." Joint European Conference on Machine Learning and Knowledge Discovery in Databases. Springer, Cham, 2021.
[2] Ming, Yifei, Ying Fan, and Yixuan Li. "Poem: Out-of-distribution detection with posterior sampling." International Conference on Machine Learning. PMLR, 2022.
[3] Bitterwolf, Julian, et al. "Breaking Down Out-of-Distribution Detection: Many Methods Based on OOD Training Data Estimate a Combination of the Same Core Quantities." International Conference on Machine Learning. PMLR, 2022.

**Summary Of The Paper:**

The paper presents a solution to the OOD detection problem. Starting from approaches that train the model on auxiliary data, the authors suggest that the distribution of auxiliary data used for training might differ from that to be encountered in practice limiting the detection capabilities. As such the authors proposed to diverse the ood data using implicit transformation via perturbing the model parameters.
Based on ReLU networks, the method uses the perturbation with to the worse regret, gradient approximation is used to limit the search space. Theoretical aspects are well supported. The method shows significant improvements over existing works on several OOD benchmarks.


**Summary Of The Review:**

The paper proposed an important solution to the OOD detection tackling the distribution gap between auxiliary ood data and real one.
The method is supported with theoretical and empirical evidence.

---

> ### Author Response · Authors · 2022-11-14
> **Response to Reviewer UUPv**
>
> We sincerely thank you for your constructive comments! Please find our responses below.
>
> > Q1: Some closely related works are missing.
>
> A1. Thank you for your kind suggestion. We will cite the mentioned papers in our revision.
>
> > Q2: Since the proposed perturbation are parameters based, it is not well discussed to which layers are the perturbations applied or the role of the used architecture. Not sure why Wide ResNet is used for Cifar experiments while ResNet 50 for ImageNet.
>
> A2. Thank you for your concern. By default, we will apply perturbation (i.e., implicit data transformation) for all layers of parameters. Its effectiveness is experimentally verified in Table 15 in Appendix C.3 (comparing with sub-model perturbation). It can be further explained by Lemma 1, in that perturbing more layers of the model can lead to more powerful data transformation, which can benefit OOD detection better. Moreover, our specification of the model architectures follows previous experimental setups [1,2]. We will make it more precise in our revision.
>
> [1] Weitang Liu, et al. "Energy-based Out-of-distribution Detection", in NeurIPS 2020.
>
> [2] Yiyou Sun, et al. "ReAct: Out-of-distribution Detection with Rectified Activations", in NeurIPS 2021.
>
> > Q3: With no code available it is untrivial to implement the proposed methods. Hyper-parameters are not discussed on how they were chosen.
>
> A3. We use validation data separated from ID and surrogate OOD data for hyper-parameter tuning, as mentioned in Section 5. Further, we release the code by the [anonymous Github repository](https://anonymous.4open.science/r/DOE/README.md). We will add a formal link if the paper is accepted.

---

> ### Author Response · Authors · 2022-11-18
> **We are looking forward to your responses or further suggestions/comments!**
>
> Dear Reviewer UUPv,
>
> We have addressed your initial concerns regarding our paper. We are happy to discuss them with you in the openreview system if you still have some concerns/questions. We also welcome new suggestions/comments from you!
>
> Best regards,
>
> Authors of #363

---

> ### Author Response · Authors · 2022-11-22
> **We are looking forward to your responses or further suggestions/comments!**
>
> Dear Reviewer UUPv,
>
> We appreciate your efforts and time in providing constructive feedback and comments. Could you check our response and confirm whether we have solved your concern? If there are further questions or suggestions, we will address them to improve our submission.
>
> Sincerely,
>
> Authors of #363

---

### Author Response · Authors · 2022-11-17
**General Author Response**

Sincerely thanks for the constructive suggestions/comments of all the reviewers. We are also glad that all the reviewers agree with our novelty, theoretical insights, and good empirical results. Here, we would like to restate two critical contributions of our proposal:

- We justify that model perturbation can lead to data transformation (cf., Section 3.1). In theorem 1, we demonstrate that the distribution of transformed data can be very different from that of the original data. In Lemma 1, we further show that the corresponding transform function is complex enough with layer-wise non-linearity, where deeper models induce stronger transformations.

- We suggest that the distributional-robust learning scheme can mitigate the OOD distribution gap (cf., Section 4.1 and Appendix B.1). By definition 1, we employ the WOR to measure the detectin capability of the detection model, which is superior to the risk counterpart. In Eq. (6), we use implicit data transformation to search for the worst OOD data, free from tedious human design and complex generative models.


We have uploaded a revised version and marked the revision in blue for the latest submission. Some major changes are:

- We improve the theoretical results in Proposition 1 and update the proof process in Appendix A.1-A.3. We change the assumption about determinants to a new assumption about eigenvalues. By applying the Jordan-Chevalley decomposition, we justify the conclusion in Proposition 1. Further, the related discussions and proofs in Appendix A.2-A.3 have been modified accordingly.

- We further explain why distribution robustness can mitigate the OOD distribution gap. Specifically, we consider two realistic situations: (1) the true OOD distribution contains all the different OOD situations, and (2) the capacity of implicit data transformation is large enough.


- We provide additional experimental results in Appendix C.1 and C.3. We demonstrate that (1) the implicit data transformation and the distributional-agnostic learning scheme mainly contribute to the improved results of DOE, and (2)  DOE can still lead to the improved results over OE for the setup of training from scratch. We also emphasize the experimental settings about backbone models and hyper-parameter tuning schemes.


We thank the reviewers again and sincerely look forward to further discussion!

---

### Decision · Program_Chairs · 2023-01-20

**Decision:**

Accept: poster

**Justification For Why Not Higher Score:**

The presentation of this submission requires some improvements.

**Justification For Why Not Lower Score:**

The paper presents some new insights into improving OOD detection via model parameter perturbation. Given the strong empirical results, the community can benefit from the presented insights.

**Metareview: Summary, Strengths And Weaknesses:**

Outlier exposure (OE) based approaches improve the out-of-distribution (OOD) detection performance by fine-tuning models on a set of surrogate OOD samples. However, these methods assume having access to an OOD set which may not cover all possible OOD scenarios at the test time. This submission suggests that one can implicitly extend the surrogate OOD set via perturbing model parameters. This observation results in a training objective in which the model is trained via a standard classification loss and a notion of worse OOD regret. On the positive side, the submission provides new insight into outlier exposure training with a model parameter perturbation approach. The empirical results show strong performance for the proposed approach. However, the connections between gradient norm and WOR are not well motivated, and it is not clear whether the resulting distribution from perturbations is indeed closer to the true distribution of OOD data. Overall, the presentation of ideas and the connections between different components could be improved and better motivated. After careful consideration, the AC believes that the merits of this submission outweigh its weaknesses, and recommends accept.

**Note From Pc:**

if the above contains the word "oral" or "spotlight" please see: "oral" presentation means -> notable-top-5% and "spotlight" means -> notable-top-25%. As stated in our emails, we are disassociating presentation type from AC recommendations

**Summary Of Ac-Reviewer Meeting:**

N/A